# T cell exhaustion and a failure in antigen presentation drive resistance to the graft-versus-leukemia effect

Meng Zhou[1], Faruk Sacirbegovic[1], Kai Zhao[1], Sarah Rosenberger[1] & Warren D. Shlomchik [1,2,3,4 ✉]

In hematopoietic cell transplants, alloreactive T cells mediate the graft-versus-leukemia (GVL) effect. However, leukemia relapse accounts for nearly half of deaths. Understanding GVL failure requires a system in which GVL-inducing T cells can be tracked. We used such a model wherein GVL is exclusively mediated by T cells that recognize the minor histocompatibility antigen H60. Here we report that GVL fails due to insufficient H60 presentation and T cell exhaustion. Leukemia-derived H60 is inefficiently cross-presented whereas direct T cell recognition of leukemia cells intensifies exhaustion. The anti-H60 response is augmented by H60-vaccination, an agonist αCD40 antibody (FGK45), and leukemia apoptosis. T cell exhaustion is marked by inhibitory molecule upregulation and the development of TOX[+] and CD39[−]TCF-1[+] cells. PD-1 blockade diminishes exhaustion and improves GVL, while blockade of Tim-3, TIGIT or LAG3 is ineffective. Of all interventions, FGK45 administration at the time of transplant is the most effective at improving memory and naïve T cell anti-H60 responses and GVL. Our studies define important causes of GVL failure and suggest strategies to overcome them.

[1] Department of Medicine, University of Pittsburgh School of Medicine, Pittsburgh, PA, USA. [2] Department of Immunology, University of Pittsburgh School of Medicine, Pittsburgh, PA, USA. [3] The Starzl Transplantation Institute, University of Pittsburgh School of Medicine, Pittsburgh, PA, USA. [4] The Hillman UPMC Cancer Center, University of Pittsburgh School of Medicine, Pittsburgh, PA, USA. ✉email: warrens@pitt.edu

Allogeneic hematopoietic stem cell transplantation (alloSCT) can cure patients with hematologic neoplasms, most commonly acute myeloblastic leukemia (AML). Much of the efficacy of alloSCT is due to alloreactive αβT cells in the donor graft, which can kill recipient leukemia cells, thereby mediating the graft-vs-leukemia (GVL) effect[1]. Unfortunately, alloreactive T cells also attack normal host (recipient) tissues, causing graft-vs-host disease (GVHD)[2–4]. When donors and recipients are MHC-matched, alloreactive T cells target minor histocompatibility antigens (miHAs), which are the peptide products of nonsynonymous polymorphisms that distinguish the donor and host[5]. Alloreactive T cells that recognize miHAs expressed by leukemia cells can mediate GVL.

Despite alloreactive T cells developing in all recipients of T cell-replete grafts, relapsed malignant disease remains the greatest single cause of post-transplant mortality[6]. Unfortunately, there has been little progress in reducing relapse, especially of myeloblastic leukemias[7,8]. A barrier to making progress has been an incomplete understanding of the biology of relapse[7].

Several nonexclusive mechanisms could contribute to GVL failure. Leukemia clones resistant to T cell killing could emerge under immune selection. This has been documented in HLA-mismatched transplants wherein relapsed leukemic cells can lose the targeted unshared HLA allele[9,10]. Relapsed leukemia samples from alloSCT recipients can have lower levels of HLA expression relative to pre-transplant specimens, and this could have been a consequence of immune selection[11,12]. HLA could be increased in relapse specimens by IFN-γ[13], consistent with mouse models[14].

Alternatively, or in addition, alloreactive T cells could fail to mount a response of sufficient magnitude and duration to completely clear or suppress leukemia cells. In one extreme, T cells could retain their intrinsic ability to be activated post-transplant but, with the elimination of miHA-bearing host hematopoietic cells, there could be insufficient miHA presentation by antigen presenting cells (APCs) to sustain them[15,16]. In the other extreme, T cells could be exhausted and unable to mount a strong response even with quality antigen presentation. Given the importance IFN-γ may have in promoting GVL against AML[11,13,14], T cell exhaustion, which results in low IFN-γ production[17], would be anticipated to diminish GVL.

To fully understand mechanisms of leukemia relapse post-alloSCT it is necessary to unequivocally identify GVL-inducing miHA-reactive T cells. This has previously not been possible in polyclonal mouse models and with human samples from alloSCT recipients because in these situations alloreactive T cells target many miHAs, most of which are unknown, and GVL-inducing T cells cannot be easily tracked. Moreover, even if all miHA-reactive T cells could be specifically identified, it would difficult to know which specificities are critical for GVL.

To address these limitations, we used a tractable mouse model wherein GVL is exclusively mediated by polyclonal alloreactive CD8 cells that target the mouse miHA H60[18]. H60 is predominantly expressed on hematopoietic cells[19], including leukemia cells, and therefore represents an ideal type of miHA that has been proposed as a clinical target[5,20,21]. H60 reactive T cells are specific in that they only mediate GVL against H60[+] and not H60[−] leukemia cells[18]. In these studies a clinically relevant model of blast crisis chronic myelogenous leukemia (BC-CML) created by co-transducing H60-expressing mouse bone marrow (BM) with retroviruses that express cDNAs encoding the human bcr-abl and NUP98/HOXA9 translocations was used[18,22]. These are bona fide oncogenes, representative of the classes of molecular drivers of AML[23]. Along with gene-modified leukemias, gene-deficient and transgenic donors and recipients, we use these tools to dissect and therapeutically address mechanisms of GVL failure. We show here that GVL fails due to insufficient antigen presentation, and the development of T cell exhaustion. The former could be improved by H60-vaccination while the effect of T cell exhaustion was mitigated by an agonist antibody to CD40 given at the time of transplant and by PD-1-blockade. Taken together these data provide new insights into GVL failure and chart a path for improving adoptive immunotherapies in the future.

## Results

**A tractable GVL system.** To create a population of trackable donor CD8 cells reactive against a miHA expressed by leukemia cells, we vaccinated C3H.SW (H-2[b]) or B6 (H-2[b]) mice against the K[b]-restricted mouse miHA H60[19] using an antibody against DEC205 which was modified to express the H60 epitope LTFNYRNL (DEC-H60) with an agonist antibody against CD40 (FGK45)[18]. CD8 memory cells ($T_M$) reactive against H60 ($T_{MH60}$) were mostly CD62L[+]CD44[+] central memory cells ($T_{CM}$) with fewer CD62L[−]CD44[+] effector memory cells ($T_{EM}$). In most experiments, B6.H60 mice (congenic for H60[18]) were irradiated and reconstituted with C3H.SW or B6 T cell-depleted BM (referred to as BM), with CD8[+]CD44[+] $T_M$ from H60-vaccinated C3H.SW or B6 donors, with or without H60[+] BC-CML[18] (referred to as BC-CML). The number of transferred CD8[+] $T_M$ was adjusted to give a defined number of $T_{MH60}$ (between 3.5 and $10 \times 10^3$), but H60 tetramer-positive (Tet[H60+]) cells were not sort-purified. While a mix of both $T_{CM}$ and effector memory $T_{EM}$ Tet[H60+] cells were transferred, most expansion was from the $T_{CM}$ Tet[H60+] cells (Supplementary Fig. 1).

**BC-CML cells outstrip the anti-H60 T cell response.** To define the kinetics of BC-CML and Tet[H60+] T cell expansion, we sacrificed cohorts 7, 14, and 21 days post-transplant in the C3H. SW→B6.H60 system. Tet[H60+] cells outnumbered BC-CML cells at day +7 and were roughly equivalent at day +14 (Fig. 1). There was no further increase in Tet[H60+] T cells after day +14, with or without BC-CML, whereas BC-CML cells continued to expand in spleen and were stable in the BM. Therefore, despite abundant antigen in the form of H60[+] BC-CML cells, the anti-miHA T cell response flattens. These data were compatible with GVL being limited by the emergence of GVL-resistant clones or by a failure in the T cell response.

**Immune selection does not account for GVL resistance.** To address whether alloimmunity selects for GVL-resistant BC-CML cells, we performed in vivo CTL assays on nonselected BC-CML and relapsed BC-CML (Fig. 2a). Relapsed BC-CML cells were harvested from B6.H60 mice that had been transplanted with BC-CML cells and $T_{MH60}$ 25-28 days prior. Nonselected BC-CML cells were harvested from B6.H60 mice transplanted with BC-CML cells without $T_{MH60}$. Nonselected (cell tracker violet-labeled), relapsed (DeepRed-labeled) and control H60[−]K[b−/−] (unlabeled) BC-CML cells were injected into B6 mice that had or had not been immunized with DEC-H60 and FGK45 seven days prior. When analyzed 18 h later, fresh and relapsed BC-CML cells were equally killed by anti-H60 effectors (Fig. 2b, c).

We also compared killing of SIINFEKL-pulsed BC-CML cells by OT-1 transgenic T cells to account for possible selection of leukemias with reduced H60 expression. Unpulsed and SIINFEKL-pulsed unselected and relapsed BC-CML cells were transferred into B6 mice that had been seeded with OT-1 cells and vaccinated with an anti-DEC205 antibody modified to express ovalbumin and FGK45 seven days prior. Again, relapsed BC-CML cells were not resistant to T cell killing relative to unselected BC-CML cells (Fig. 2d, e). These data indicate that in our model immune selection does not account for GVL

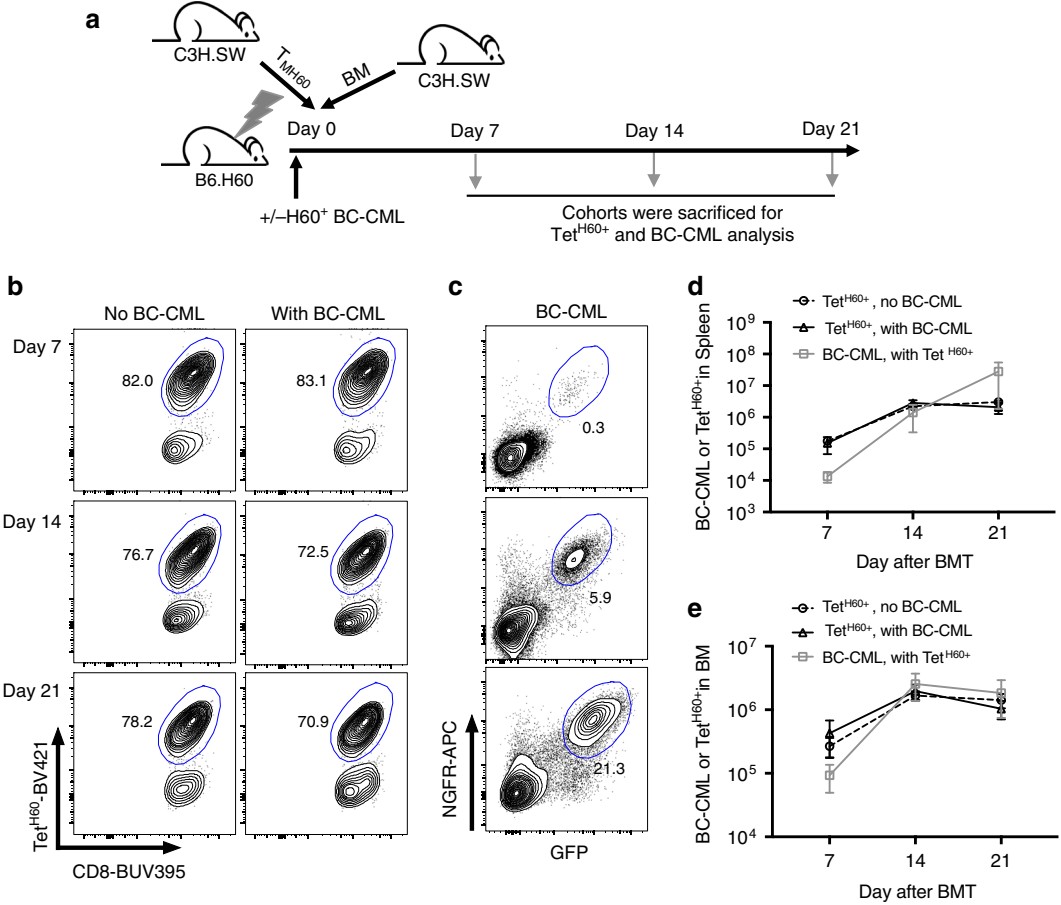

**Fig. 1 BC-CML cells survive despite a robust anti-H60 CD8 response. a** Experimental design. B6.H60 mice were irradiated and reconstituted with C3H. SW BM and $T_{MH60}$ (containing $10^4$ CD8$^+$Tet$^{H60+}$ cells), with or without B6.H60 BC-CML. Cohorts were sacrificed at days 7, 14, and 21 post-transplantation for enumeration of BC-CML and Tet$^{H60+}$ cells in spleen and BM. Representative tetramer staining for H60 (Tet$^{H60}$) and NGFR (linked to bcr-abl) and GFP (linked to NUP98/HOXA9) expression are in **b** and **c**, respectively. Tet$^{H60+}$ and BC-CML cells were enumerated in spleen (**d**) and BM (**e**). Panels (**d**) and (**e**) are combined from two repetitions ($n = 8$ per group). Bars are mean values ± SD.

resistance. We therefore further investigated the anti-H60 CD8 response.

**Ineffective antigen presentation limits GVL.** Because H60 is mostly expressed by hematopoietic cells, effective antigen presentation may become limiting as host hematopoietic cells die from radiation and T cell killing[15]. Conversely, H60 would become increasingly available from expanding BC-CML cells; however, how well BC-CML cells directly stimulate T cells or whether their miHAs are effectively cross-presented by donor-derived cells was unknown. To measure antigen presentation at day 14 post-transplant, (when Tet$^{H60+}$ cells peak; Fig. 1), we infused congenic CD45.1$^+$ C3H.SW $T_{MH60}$ 14 days after B6.H60 mice were transplanted with C3H.SW BM and CD45.2$^+$ $T_{MH60}$, with or without BC-CML. As a positive control for the capacity of day 14 (D14) $T_{MH60}$ to respond to H60, some mice were also immunized with DEC-H60 and FGK45 (design, Fig. 3a). By day +21, CD45.1$^+$ D14 $T_{MH60}$ had undergone little expansion, nowhere near that of $T_{MH60}$ infused on day 0 (Fig. 3b, c). Immunization with FGK45 + DEC-H60 increased D14 CD45.1$^+$Tet$^{H60+}$ progeny more than 10-fold (with or without BC-CML) indicating that D14 cells can be activated with effective antigen presentation. Importantly, D14 $T_{MH60}$ combined with FGK45 + DEC-H60 reduced BC-CML numbers in spleen and BM relative to infusion of only $T_{MH60}$ (Fig. 3d).

Surprisingly, FGK45 alone augmented the anti-H60 response, even in the absence of BC-CML, indicating that there were sources of H60 not being effectively presented (Fig. 3g, h). FGK45 combined with D14 CD45.1$^+$ $T_{MH60}$ also improved GVL (Supplementary Fig. 2A). GVL promotion by FGK45 was not due to a direct action on BC-CML cells as FGK45 did not reduce BC-CML numbers in RAG2$^{-/-}$γc$^{-/-}$ mice transplanted with BC-CML cells and no T cells (Supplementary Fig. 2B).

The addition of DEC-H60 to FGK45 consistently led to a greater increase of D14-derived Tet$^{H60+}$ cells in BM and expansion of day 0 (D0)-derived Tet$^{H60+}$ cells enumerated on day +21 (Fig. 3g). We therefore performed experiments wherein transplanted mice received D14 $T_{MH60}$ alone, with DEC-H60 or with DEC-H60 + FGK45 (Supplementary Fig. 2C). DEC-H60 alone augmented expansion of D14 $T_{MH60}$, though not as effectively as FGK45 + DEC-H60, indicating that both antigen and suboptimal APC activation limit H60-reactive T cell expansion.

To determine whether FGK45 was acting on donor and/or residual host-derived APCs we compared its effect in mice transplanted with wild type (wt) or CD40$^{-/-}$ BM using the B6→B6.H60 system as C3H.SW CD40$^{-/-}$ mice were not available. Irradiated B6.H60 mice were reconstituted with B6 or B6 CD40$^{-/-}$ BM. On day +14 all mice received B6 CD45.1 $T_{MH60}$ and a cohort from each group was treated with FGK45. FGK45 had less of an impact on Tet$^{H60+}$ expansion in recipients of B6 CD40$^{-/-}$ BM,

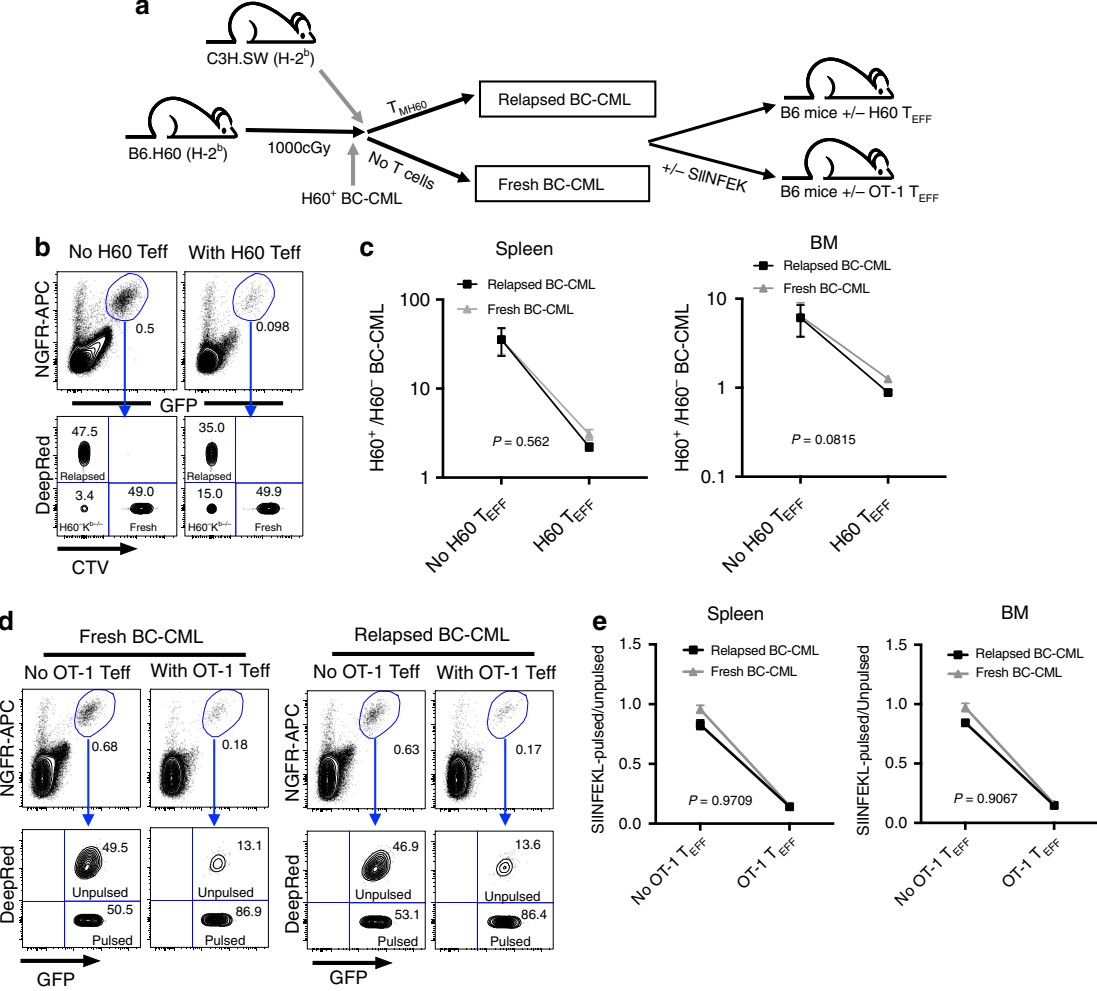

**Fig. 2 Progressive leukemia is not due to emergence of GVL-resistant leukemia cells.** Unselected (fresh) BC-CML and GVL-relapsed BC-CML were prepared as described in the methods. Experimental scheme (**a**). To test killing directed against H60, fresh and relapsed BC-CML were labeled with CellTrace Violet (CTV) or CellTracker DeepRed (respectively) and transferred together with $H60^-K^{b-/-}$ BC-CML (unlabeled) to B6 mice that did or did not have anti-H60 $CD8^+$ $T_{eff}$ generated by vaccination. Eighteen hours later, mice were sacrificed to enumerate BC-CML cells in spleen and BM. Show are representative flow cytometry of splenocytes from mice without and with H60 $T_{eff}$ (**b**) and the ratios of fresh or GVL-relapsed BC-CML to $H60^-K^{b-/-}$ BC-CML in spleen and BM (**c**). **d**, **e** Fresh and relapsed BC-CML cells that were either unpulsed or pulsed with SIINFEKL (DeepRed-labeled) were injected into OT-1-seeded or unmanipulated mice. Shown are representative flow cytometry of splenocytes (**d**) and the ratios of SIINFEKL-pulsed/unpulsed in spleen and BM (**e**). **b**–**e** are representative of two experiments with five mice per group per experiment. A paired two-sided $t$-test was used for statistical analysis (mean ± SD). $P = 0.562$ and $P = 0.0815$ comparing the ratios of fresh or GVL-relapsed BC-CML to $H60^-K^{b-/-}$ BC-CML in spleen and BM, respectively. $P = 0.9709$ and $P = 0.9067$ comparing the ratios of SIINFEKL-pulsed/unpulsed in spleen and BM, respectively.

indicating that FGK45's greatest activity is on donor APCs cross-presenting host H60 (Supplementary Fig. 2D). Because mice did not receive day 0 $T_{MH60}$ there was likely a substantial number of residual recipient APCs, which may explain the activity of FGK45 in recipients of $CD40^{-/-}$ BM.

To determine whether donor dendritic cells (DCs) were essential, B6.H60 mice were transplanted with wt B6 CD45.2 $T_{MH60}$ (to enhance the clearance of recipient APCs) and BM from B6 mice that express the diphtheria toxin receptor (DTR) under control of the CD11c gene (CD11c-DTR[24]) or B6 $CD40^{-/-}$ BM. Beginning on day +10, a cohort of CD11c-DTR BM recipients was injected with diphtheria toxin (DT) every other day to deplete donor-derived DCs. On day +14, all mice received B6 $CD45.1^+$ $T_{MH60}$, with or without FGK45 (design, Supplementary Fig. 2E). FGK45 was again less effective in recipients of B6 $CD40^{-/-}$ BM (Fig. 3i). Importantly, donor DC-depletion completely prevented FGK45's augmentation of the $Tet^{H60}$

response, indicating that donor DCs that cross-present recipient H60 were the major FGK45 targets (Fig. 3i).

**Effect of leukemia-derived H60 on the anti-H60 response.** It was surprising that insufficient H60 presentation limited the activation of H60-reactive T cells in mice with a substantial burden of $H60^+$BC-CML. To test how efficiently leukemia-derived miHAs are cross-presented, irradiated B6 mice were reconstituted with C3H.SW BM, C3H.SW $T_{MH60}$ and B6 $H60^+K^{b-/-}$ BC-CML cells. At day +13 post transplantation, $Tet^{H60+}$ cells were few relative to similarly transplanted B6.H60 recipients and similar to control mice transplanted with $H60^-K^{b-/-}$ BC-CML cells (Fig. 4a), indicative of inefficient cross-presentation.

To determine how well leukemia cells directly stimulate alloreactive T cells, irradiated B6 beta-2-microglobulin-deficient ($β2M^{-/-}$) mice were reconstituted with B6 $β2M^{-/-}$ BM, wt $H60^+$ BC-CML, and B6 $T_{MH60}$. In such mice only BC-CML cells

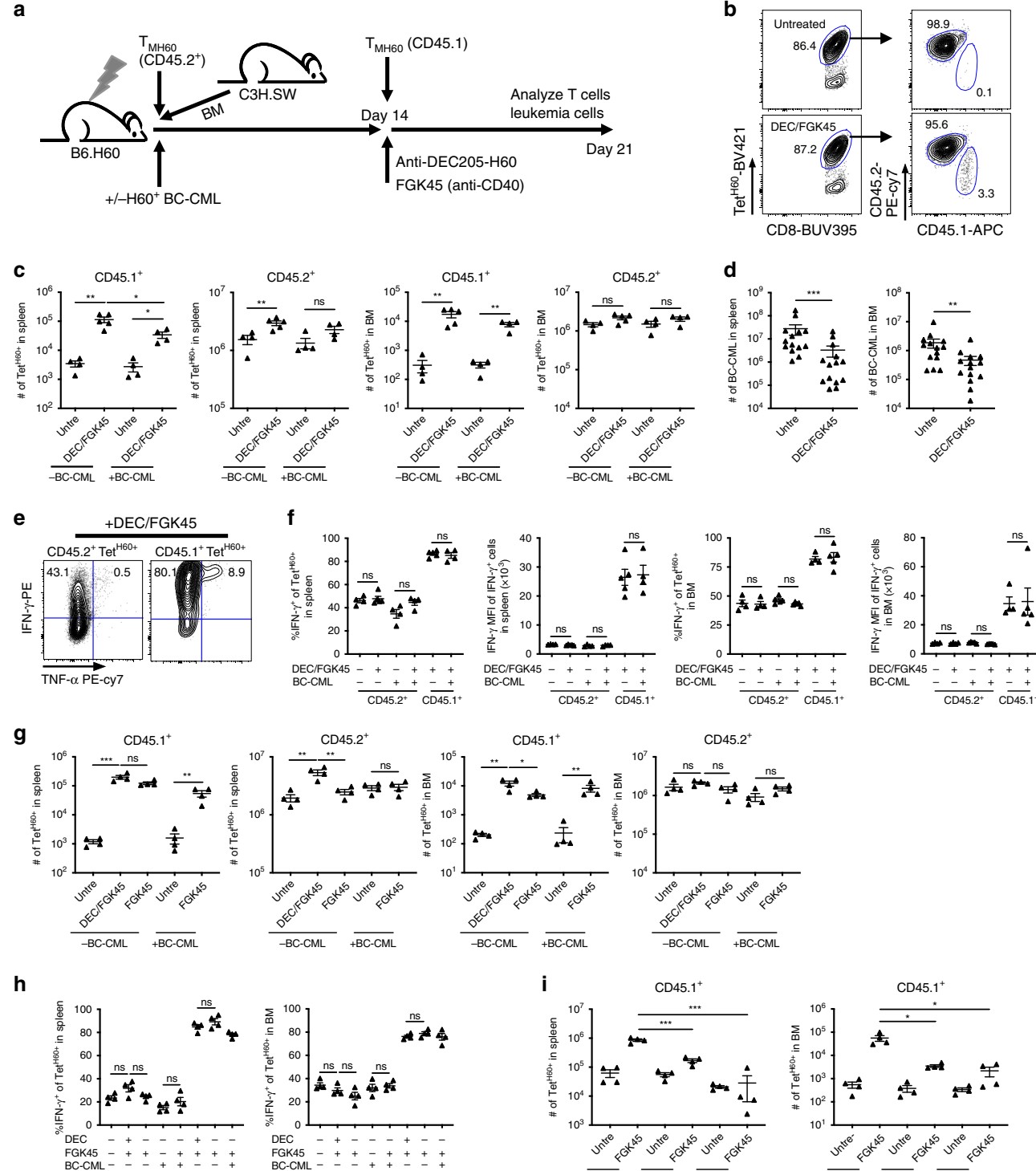

were MHCI$^+$ and capable of stimulating $T_{MH60}$. At day +10 post-transplant we could not detect CD8$^+$Tet$^{H60+}$ cells whereas there was a large Tet$^{H60+}$ response in control B6.H60 recipients (Fig. 4b).

To test whether leukemia cells need to be killed for their miHAs to be cross-presented we transplanted B6 mice with CD45.2$^+$ C3H.SW BM, K$^{b-/-}$H60$^+$ BC-CML along with C3H.SW CD4 cells, which mediate GVL in this model[25]. On day +14, mice received CD45.1$^+$ $T_{MH60}$ and FGK45. Despite significant CD4-mediated GVL, there was little Tet$^{H60+}$ expansion (Fig. 4c).

It was possible that the magnitude or timing of CD4$^+$ T cell killing was suboptimal for effective cross-presentation. We

therefore engineered H60$^+$K$^{b-/-}$ BC-CML cells to express an inducible caspase 9 (iCasp9)[26]. Treatment of mice harboring these leukemias with AP20187 (chemical induced dimerization; CID) rapidly induced apoptosis (Supplementary Fig. 3). Irradiated B6 or B6 K$^{b-/-}$ mice (to restrict H60 presentation to donor cells) were reconstituted with C3H.SW BM and H60$^+$K$^{b-/-}$ iCasp9 BC-CML. Twelve to 14 days post-transplant, a cohort was injected with CID. On day +14, all mice received C3H.SW $T_{MH60}$ and FGK45 (design, Fig. 4d) to measure cross-presentation. On day +21 there was robust Tet$^{H60+}$ T cell expansion in CID-treated B6 and B6 K$^{b-/-}$ recipients and significant expansion even in untreated mice, likely due to

**Fig. 3 $T_{MH60}$ given at time of transplant fail due to a lack of effective antigen presentation.** Experimental scheme (**a**). B6.H60 mice were irradiated and reconstituted with C3H.SW BM, C3H.SW CD45.2$^+$ $T_{MH60}$ (containing $10^4$ Tet$^{H60+}$ cells) with or without BC-CML. At day 14 post-transplant, fresh C3H.SW CD45.1$^+$ $T_{MH60}$ (containing $10^4$ Tet$^{H60+}$ cells) were infused. Some mice also received DEC-H60 + FGK45 to further activate H60-reactive T cells. Mice were sacrificed on day 21 post-transplant for analysis of BC-CML and $T_{MH60}$ progeny in spleen and BM. **b** Representative Tet$^{H60}$ staining of splenocytes at day 21. Shown are Tet$^{H60}$ cells from day 0- (CD45.2$^+$) and day 14-derived (CD45.1$^+$) $T_{MH60}$. Quantification of day 0 and day 14 $T_{MH60}$ progeny in spleen and BM are in **c**. Total numbers of BC-CML in spleen and BM are in **d**. Representative flow cytometry (**e**) and quantification of IFN-γ expression (**f**) of D14 CD45.1$^+$ and D0 CD45.2$^+$ Tet$^{H60+}$ cells harvested at day +21. Data are pooled from 3 independent experiments with 14 mice per group (**d**), or representative of 3 independent experiments (4–5 mice per group) with similar results (**b**, **c**, **f**). **g**, **h** Mice were transplanted as in panel A. Fourteen days post-transplant, mice were injected with fresh C3H.SW CD45.1 $T_{MH60}$ with DEC-H60 and FGK45 or FGK45 alone. The total number of day 0 $T_{MH60}$ progeny (CD45.2$^+$) and day 14 $T_{MH60}$ progeny (CD45.1$^+$) in spleen and BM were quantitated (**g**), and the percentages of these that are IFN-γ+ are shown in **h**. Data are representative of two independent experiments with 4 mice per group. **i** B6.H60 mice were irradiated and reconstituted with B6 (CD45.2$^+$) $T_{MH60}$ (containing $10^4$ Tet$^{H60+}$ cells) and donor BM from B6 CD11c-DTR or B6 CD40$^{-/-}$ mice. At day +14, $5 \times 10^4$ fresh B6 CD45.1$^+$ $T_{MH60}$ were infused, with or without FGK45. A cohort of CD11c-DTR BM recipients was injected with DT every other day from day 10 to day 20. Mice were sacrificed at day +21 and $T_{MH60}$ progeny in spleen and BM were quantitated ($n = 4$ per group). For all panels, data were analyzed by an unpaired Student two-sided $t$-test. Bars are mean values ± SEM, *$P < 0.05$; **$P < 0.01$; ***$P < 0.001$; and ns not significant.

spontaneous iCasp9 dimerization (Fig. 4e). Therefore, leukemia-derived antigens can be cross-presented by donor-derived APCs, but only after substantial leukemia apoptosis.

**T cell exhaustion contributes to GVL failure**. While DEC-H60 and FGK45 induced expansion of $T_{MH60}$ infused on day +14, they less effectively stimulated the progeny of $T_{MH60}$ infused on day 0 (Fig. 3c, g). Moreover, a smaller fraction of day 0 $T_{MH60}$ progeny produced IFN-γ with peptide restimulation (Fig. 3e, f, h), characteristic of T cell exhaustion. We therefore further analyzed $T_{MH60}$ progeny from B6.H60 mice transplanted with C3H.SW BM and $T_{MH60}$, with or without H60$^+$ BC-CML. By days +21–25, Tet$^{H60+}$ cells from BM and spleen had uniformly upregulated PD-1 and most progeny expressed Tim-3, TIGIT and LAG3, all associated with exhaustion (Fig. 5a, b and Supplementary Fig. 4A). They also had reduced IFN-γ production relative to fresh $T_{MH60}$ and to Tet$^{H60+}$ cells harvested at day +7 post-transplant (Fig. 5c and Supplementary Fig. 4B, C).

Eomes and Blimp-1 were high in Tet$^{H60+}$ cells, again consistent with an exhaustion phenotype (Fig. 5d). Importantly, relative to mice transplanted without leukemia, Tet$^{H60+}$ cells from mice transplanted with BC-CML had more characteristics of exhaustion–increased inhibitory molecule expression, increased Blimp-1 and Eomes and reduced IFN-γ production–indicating that rather than being stimulatory, BC-CML cells augment exhaustion (Fig. 5a–d). This effect required cognate TCR:MHCI contact as K$^{b-/-}$ H60$^+$BC-CML did not intensify exhaustion (Fig. 5e). H60$^+$PD-L1/L2$^{-/-}$ BC-CML induced similar exhaustion as did H60$^+$PD-ligand-intact BC-CML indicating that antigen exposure and not PD-ligands promote exhaustion (Fig. 5f).

Tet$^{H60+}$ cells isolated post transplantation had metabolic characteristics of exhaustion[27]. Freshly isolated $T_{MH60}$ had low-level 2-NBDG uptake and low mitochondrial mass and mitochondrial membrane potential as expected for quiescent cells (Supplementary Fig. 4D, E). In contrast, on day +7 post transplantation, splenic and BM Tet$^{H60+}$ cells had increased glucose uptake and mitochondrial mass, with a majority of cells also having bright TMRE staining, consistent with active respiration (Supplementary Fig. 4D, E). By day +21, however, Tet$^{H60+}$ cells had lost mitochondrial mass, with few being TMRE-bright (Fig. 5g and Supplementary Fig. 4D, E). The presence of BC-CML led to a modest increase in mitochondrial mass, but without a consistent change in mitochondrial depolarization (Fig. 5g).

**Inhibition of PD-1 but not Tim-3, LAG3 or TIGIT, improves GVL.** Given the evidence for T cell exhaustion, we investigated

whether post-transplant PD-1 blockade would improve $T_{MH60}$ performance. Irradiated B6.H60 mice were reconstituted with C3H.SW BM, C3H.SW $T_{MH60}$, and B6.H60 BC-CML. α-PD1 or an isotype control was begun on day +7 to avoid adverse effects of PD-1 blockade on early T cell activation[28]. α-PD1 reduced the percentage of BC-CML cells in blood on day +14 and their numbers in spleen and BM on day +21. α-PD1 also increased the number of Tet$^{H60+}$ cells in BM (Fig. 6a, b). Importantly, in α-PD1 treated mice Tet$^{H60+}$ cells produced more IFN-γ and had a reduced expression of TIGIT (Fig. 6b–d). The increase in IFN-γ production with in vitro stimulation reflected increased in vivo IFN-γ production as MHCII expression on BC-CML cells, which is IFN-γ-regulated[14], was higher with PD-1-blockade (Fig. 6e).

We next tested blockade of Tim-3, TIGIT and LAG3, alone or in combination with α-PD1 (design, Supplementary Fig. 5A). In all experiments α-PD1 improved GVL, increased the number of BM Tet$^{H60+}$ cells, the percentage of IFN-γ$^+$Tet$^{H60+}$ cells in both BM and spleen, and decreased expression of Tim-3 and TIGIT (Fig. 6f–i; Supplementary Fig. 5). However, blockade of LAG3 (Fig. 6f), Tim-3 or TIGIT (Supplementary Fig. 5B, F) did not augment GVL, even in combination with α-PD1 under conditions in which α-PD1 was only partially effective. TIGIT-blockade diminished the number of Tet$^{H60+}$ cells while increasing the percentage that produced IFN-γ (Supplementary Fig. 5G, H), perhaps by depleting cells with the highest TIGIT expression. LAG3-blockade increased the percentage of Tet$^{H60+}$ cells that produced IFN-γ but did not blunt the upregulation of Tim-3 or TIGIT (Fig. 6h, i).

**α-CD40 at time of transplantation improves GVL.** Given how effectively FGK45 promoted the activation of $T_{MH60}$ infused on day +14, we tested FGK45 in combination with $T_{MH60}$ infused on day 0 in the C3H.SW→B6.H60 system. Day 0 FGK45 dramatically reduced BC-CML numbers in spleen and BM at day +21, coincident with an increase in the number of BM Tet$^{H60+}$ cells (Fig. 7a, b). FGK45 increased IFN-γ production by Tet$^{H60+}$ cells with fewer expressing high levels of TIGIT, PD-1 and Tim-3 (Fig. 7c, d). Consistent with the increase in IFN-γ$^+$ cells, MHCII expression was higher in BC-CML cells harvested from FGK45-treated mice (Fig. 7e).

FGK45 primarily acted on recipient cells as there was no reduction in GVL in recipients of CD40$^{-/-}$ BM (Fig. 7f). It was possible that FGK45 increased direct presentation of K$^b$:LTFNYRNL; alternatively or in addition it could have improved cross-presentation. To address this, we made mixed BM chimeras in which recipient APCs could directly present H60 ([B6.H60 + B6]→B6 chimeras) or only cross-present H60 ([B6.H60 K$^{b-/-}$ + B6]→B6 chimeras). After 8 weeks, these chimeras

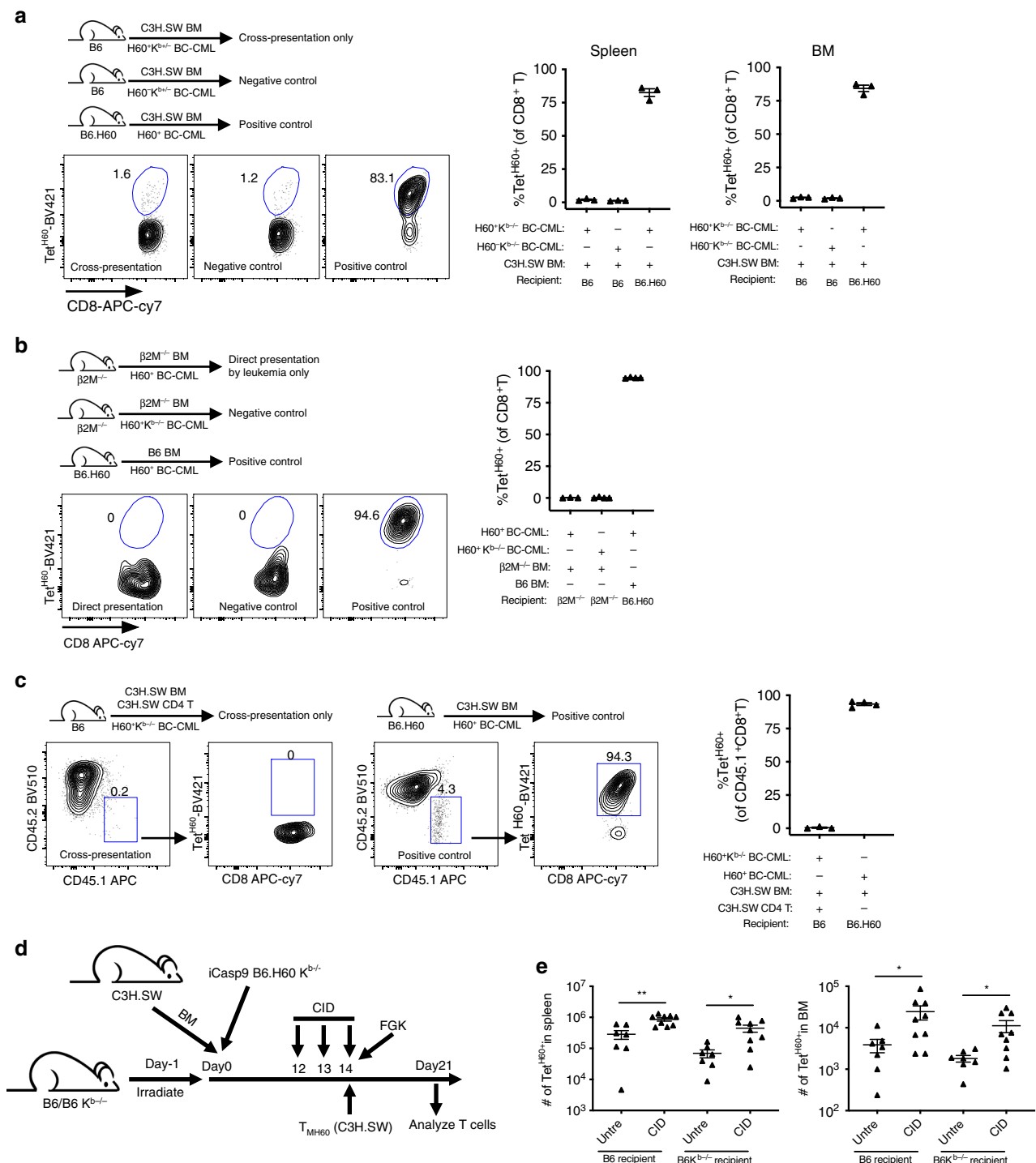

were reirradiated and retransplanted with B6 CD40$^{-/-}$ BM and B6 T$_{MH60}$, with or without FGK45 (design, Supplementary Fig. 6A). FGK45 only augmented the Tet$^{H60+}$ response in the (B6.H60 + B6)→B6 chimeras in which H60 was directly presented whereas there was no effect in (B6.H60 K$^{b-/-}$ + B6)→B6 chimeras wherein FGK45 could only promote H60 cross-presentation by B6 K$^b$-intact APCs (Fig. 7g). Nonetheless, Tet$^{H60+}$ cells ultimately expanded even in the (B6.H60 K$^{b-/-}$ + B6)→B6 chimeras, confirming that cross-presentation does take place[16]. With only H60 cross-presentation, markers of exhaustion were reduced, suggesting that the intensity of miHA exposure drives exhaustion (Supplementary Fig. 6B, C).

Although recipient DCs are not required for GVHD in models wherein host hematopoietic cells are essential[29,30], it was possible that FGK45's activity was reliant on DCs. To test this we first crossed B6.H60 mice to B6 CD11c-DTR mice[24] and then made B6.H60 CD11c-DTR→B6 K$^{b-/-}$ BM chimeras in which the only host cells that can present H60 are derived from the B6.H60 CD11c-DTR BM. These chimeras were DT- or PBS-treated, reirradiated, and transplanted with C3H.SW BM and T$_{MH60}$, with or without FGK45. FGK45 augmented the Tet$^{H60}$ response in both PBS- and DT-treated mice (Fig. 7h). Nonetheless, DT-treated mice that received FGK45 generated modestly fewer Tet$^{H60+}$ cells, suggesting that DCs, though not required, are FGK45 targets.

**Fig. 4 Leukemia apoptosis promotes miHA cross-presentation. a–c** Panels show experimental designs, representative flow cytometry and quantitation of $Tet^{H60+}$ cells. **a** To test whether BC-CML-derived H60 can be cross-presented, B6 mice were irradiated and reconstituted with C3H.SW BM, $H60^+K^{b-/-}$ BC-CML and C3H.SW $T_{MH60}$. As a positive control for $Tet^{H60+}$ expansion, irradiated B6.H60 mice were reconstituted with $H60^+$ BC-CML with BM and $T_{MH60}$ from C3H.SW or B6 mice. As a negative control, irradiated B6 mice were reconstituted with C3H.SW BM, $H60^-K^{b-/-}$ BC-CML and C3H.SW $T_{MH60}$. At day +13, few $Tet^{H60+}$ cells could be detected in recipients of $H60^+K^{b-/-}$ BC-CML, and their number was similar to that in recipients of $H60^-K^{b-/-}$ BC-CML. (**b**) To determine whether the $H60^+$ BC-CML can directly stimulated $T_{MH60}$ in vivo, irradiated B6 $β2M^{-/-}$ were reconstituted with B6 $β2M^{-/-}$ BM, B6 $T_{MH60}$ and $H60^+K^{b+/+}$ BC-CML or $H60^+K^{b-/-}$ BC-CML as a negative control. As a positive control, irradiated B6.H60 mice were reconstituted with B6 BM, B6 $T_{MH60}$ and $H60^+$ BC-CML cells. At day +10, $Tet^{H60+}$ cells were not detectable in blood in both $K^{b+/+}$ and $K^{b-/-}$ $H60^+$ BC-CML recipients. **c** To test whether GVL can increase cross-presentation, irradiated B6 mice were reconstituted with C3H.SW BM, $CD45.2^+$ C3H.SW CD4 cells and $H60^+K^{b-/-}$ BC-CML cells. At day +14 post transplant, $CD45.1^+$ C3H.SW $CD8^+$ $T_{MH60}$ cells were transferred to recipients with FGK45 to promote cross-presentation. Mice were sacrificed on day +21 and $Tet^{H60+}CD45.1^+$ cells were enumerated. As a positive control, irradiated B6.H60 were reconstituted with C3H.SW BM and $K^{b+/+}H60^+$ BC-CML with C3H.SW $CD45.1^+$ $T_{MH60}$ cells and FGK45 infused on day +14. For **a–c**, data are representative of two independent experiments with 3–4 mice per group. **d, e** Design (**d**). B6 and B6 $K^{b-/-}$ mice were irradiated and reconstituted with C3H.SW BM and iCasp9 $B6.H60K^{b-/-}$ BC-CML. CID was given on days +12–14 to induce BC-CML apoptosis. On day +14, C3H.SW $CD45.1^+$ $T_{MH60}$ cells were injected with FGK45. Mice were sacrificed on day +21 and $Tet^{H60+}$ cells in spleen and BM were quantitated (**e**). Data are combined from two experiments (7–9 mice per group). For all panels, data were analyzed by an unpaired Student two-sided $t$-test. Bars are mean values ± SEM. *$P < 0.05$; **$P < 0.01$.

While an ideal target miHA would be hematopoietically-restricted, as is H60, some miHAs may also be expressed in other tissues. We previously showed that $T_{MH60}$ cause little GVHD even in B6 actH60 mice, which express H60 ubiquitously driven by an actin promoter[18]. We therefore explored the impact of FGK45 on $T_{MH60}$ and on GVL in actH60 recipients. Irradiated actH60 mice were reconstituted with C3H.SW BM, C3H.SW $T_{MH60}$ and B6.H60 BC-CML. One group was treated with FGK45 on day 0 and mice were sacrificed on day 18. FGK45 dramatically reduced the number of BC-CML cells in BM and spleen (Fig. 7i). While the number of $Tet^{H60+}$ cells was not increased, there was an increase in the $IFN-γ^+$ fraction in spleen and a reduction of TIGIT expression in both spleen and BM, indicative of improved T cell fitness (Fig. 7j–l).

We next focused on how day 0 FGK45 alters $Tet^{H60+}$ T cell activation (design, Supplementary Fig. 7A). By day +3, $Tet^{H60+}$ cells from FGK45-treated mice had increased expression of CD25, 4-1BB, GITR and OX40 (Fig. 8a). At day +4, a greater fraction had diluted CFSE (Fig. 8b). However, a greater fraction of $Tet^{H60+}$ cells from FGK45-treated mice were annexin $V^+$ and $7-AAD^+$ (Fig. 8c), suggesting that increased proliferation primarily drives the FGK45 effect. By days +7 and +14, FGK45 treatment yielded a 5-10-fold increase in $Tet^{H60+}$ cells with no change in their $IFN-γ$ production (Fig. 8d). However, $Tet^{H60+}$ cells from FGK45-treated mice had lower expression of PD-1, Tim-3 and TIGIT, with an increase in $Eomes^{low}Tbet^{high}$ cells (Fig. 8e, f and Supplementary Fig. 7B, C) that have been reported to have a greater proliferative capacity[31].

**Exhaustion of naïve alloreactive progeny.** We used donor $T_{MH60}$ in our experiments so as we could track all GVL-inducing T cells. While $T_{CM}$ share much with naïve T cells ($T_N$), we also explored exhaustion and the impact of FGK45 on H60-reactive progeny of $CD8^+$ $T_N$. Irradiated B6.H60 mice were reconstituted with C3H.SW BM and CD8 cells from unvaccinated donors. One group received FGK45 on day 0 and mice were sacrificed on days +7 and +14 for analysis. Although we did not sort $T_N$, spontaneous $T_M$ do not mount an anti-H60 response[18]; therefore, all $Tet^{H60+}$ cells were derived from $T_N$. By day +7, FGK45 increased $Tet^{H60+}$ cells in spleen and BM by ~100-fold (Fig. 9a, b). By day +14, there were more $Tet^{H60+}$ cells in spleen but not in BM (Fig. 9c). FGK45 reduced, but did not prevent, alloreactive T cell exhaustion. In spleen, $IFN-γ$ production was similarly low in both groups; however, at day +14 more $IFN-γ^+Tet^{H60+}$ cells were present in the BM of FGK45-treated mice. Consistent with this,

FGK45 reduced the expression of PD-1, TIGIT and Eomes on $Tet^{H60+}$ cells.

FGK45 also had a strong effect on $Tet^{H60-}$ cells in recipients of CD8 cells from unmanipulated C3H.SW donors (Supplementary Fig. 8). At day +7, FGK45 had increased the number of $Tet^{H60-}$ CD8 cells in both BM and spleen more than 10-fold, coincident with a large increase in the fraction of cells that expressed PD-1, TIGIT and Tim-3. By day +14, the numbers of $Tet^{H60-}$ CD8 cells were more similar in FGK45-treated and untreated mice as were the frequencies of cells expressing PD-1, TIGIT and Tim-3. Taken in the context of the increase of H60-reactive cells in the same FGK45-treated mice, these data suggest that FGK45 also accelerated the proliferation of donor CD8 cells reactive against miHAs other than H60.

FGK45 also promoted GVL mediated by H60-reactive progeny of $CD8^+$ $T_N$. To assure that only H60-reactive T cells mediated GVL we used the B6→B6.H60 system. Irradiated B6.H60 mice were reconstituted with B6 BM and CD8 cells and B6.H60 BC-CML, with or without FGK45. FGK45 reduced the number of BC-CML cells and increased the number of $Tet^{H60+}$ CD8 cells (Fig. 9d). FGK45 decreased the expression of TIGIT and Eomes on $Tet^{H60+}$ cells, suggestive of improved T cell fitness. While the fractions of $IFN-γ^+Tet^{H60+}$ cells were similar in untreated and FGK45-treated mice, BC-CML MHCII expression was higher in FGK45-treated mice, indicative of greater in vivo $IFN-γ$ exposure.

FGK45 similarly improved GVL mediated by CD8 cells responsive against miHAs other than H60. Irradiated B6 mice were reconstituted with B6 BC-CML and C3H.SW BM and CD8 cells. One group was treated with FGK45 on day 0. At day +18, FGK45 reduced BC-CML cells in spleen and BM by ~100-fold. While we could not track CD8 cells targeting a specific miHA, FGK45 increased the $IFN-γ^+$ fraction of CD8 cells and reduced the frequency of CD8 cells with an exhaustion phenotype (Fig. 9e).

**$TOX^+$ and $TCF-1^+$ alloreactive T cells develop post-transplant.** We also explored whether the sustained availability of alloantigen in the transplant environment induces expression of TOX, a transcription factor recently shown to contribute to T cell exhaustion[32–37]. We further investigated whether a $TCF-1^+$ subpopulation of $Tet^{H60+}$ cells emerged. Such cells have been proposed as precursors of exhausted T cells[32,34–39].

Irradiated B6.H60 mice were reconstituted with C3H.SW BM and CD8 cells from either unmanipulated C3H.SW donors (referred to as $T_N$) or $T_M$ from H60-vaccinated C3H.SW mice ($T_N$). Cohorts were also injected with FGK45. TOX was

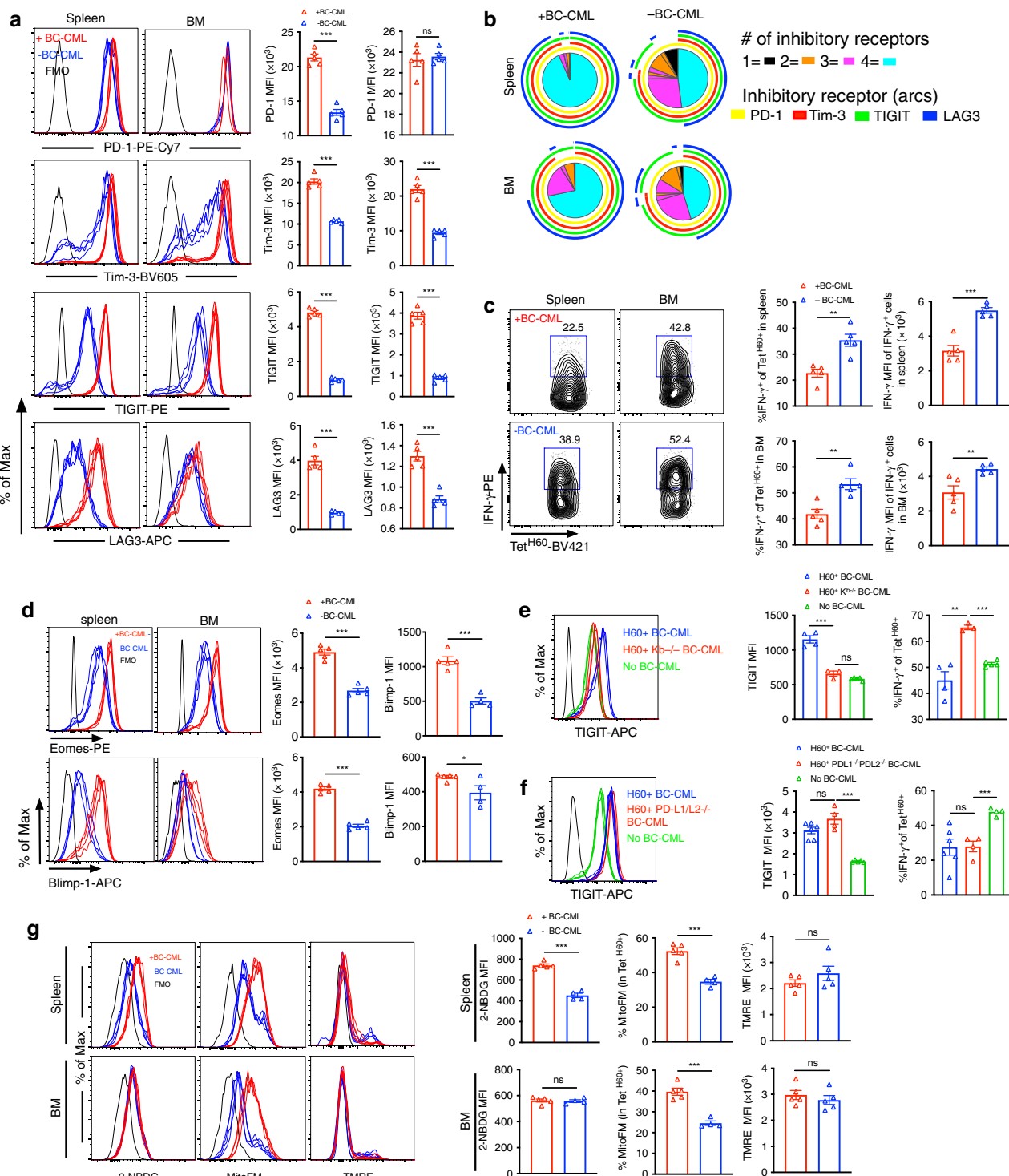

**Fig. 5 H60+ BC-CML cells contribute to exhaustion through cognate interactions. a–d** B6.H60 mice were irradiated and reconstituted with C3H.SW BM, C3H.SW T$_{MH60}$, with or without B6.H60 BC-CML. Mice were sacrificed 21–25 days post-transplantation for analysis. The expression of PD-1, Tim-3, TIGIT, and LAG-3 on Tet$^{H60+}$ cells in spleen and BM and the fraction of Tet$^{H60+}$ cells expressing these inhibitory receptors alone and in combination are in **a** and **b**, respectively. **c** Representative flow cytometry (left panels) and quantitation of IFN-γ expression (right panels). **d** Eomes and Blimp-1 expression in Tet$^{H60+}$ cells from spleen and BM. Panels **a–d** are representative of three experiments with five mice per group. **e, f** B6.H60 mice were irradiated and reconstituted with C3H.SW BM, C3H.SW T$_{MH60}$, with no BC-CML (five mice per group in **e**; four mice per group in **f**), H60+ BC-CML (four mice per group in panels **e**; six mice per group in **f**), H60+K$^{b−/−}$ BC-CML (three mice per group in **e**, **f**) or H60+PD-L1$^{−/−}$PD-L2$^{−/−}$ BC-CML (four mice per group in **e** and **f**). Mice were sacrificed at day +16 (for H60+K$^{b−/−}$ BC-CML) and day +27 (for H60+PD-L1$^{−/−}$PD-L2$^{−/−}$ BC-CML). Tet$^{H60+}$ cells were analyzed for TIGIT expression and IFN-γ production. Data are representative of two experiments. **g** Mice were transplanted as in **a–d**. On day 21 post-transplantation Tet$^{H60+}$ cells were analyzed for 2-NBDG uptake and MitoFM and TMRE staining. Shown are representative flow cytometry and mean fluorescent intensities (MFI). Data are representative of three experiments (4–5 mice per group). For all panels, data were analyzed by an unpaired Student two-sided $t$-test. Bars are mean values ± SEM. *$P < 0.05$; **$P < 0.01$; ***$P < 0.001$.

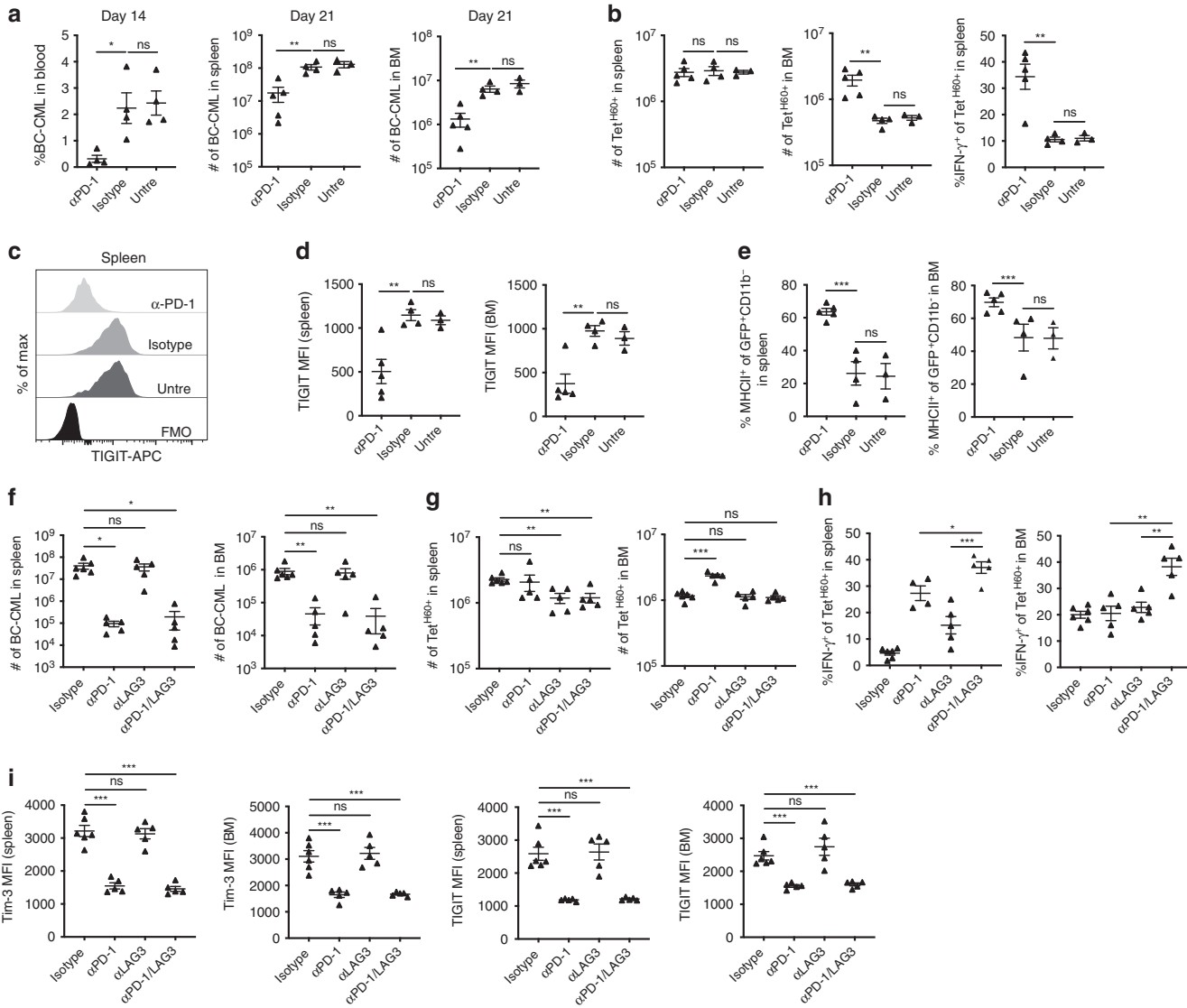

**Fig. 6 Anti-PD-1 mAb treatment augments T$_{MH60}$-mediated GVL.** B6.H60 mice were irradiated and reconstituted with C3H.SW BM, B6.H60 BC-CML, and C3H.SW T$_{MH60}$. From days +7 to +19 post transplantation, mice were treated with α-PD1 or an isotype control. Mice were sacrificed on day +21 for analysis of BC-CML and Tet$^{H60+}$ cells. The percentages of BC-CML in blood and the total number of BC-CML cells in spleen and BM at day +21 are shown in **a**. Total numbers of Tet$^{H60+}$ in spleen and BM, and the percentage of these that are IFN-γ$^+$ in spleen are in **b**. **c**, **d** Representative flow histograms (**c**) and quantification of TIGIT expression on Tet$^{H60+}$ cells in spleen and BM (**d**). (**e**) The percentages of BC-CML cells that are MHCII$^+$. (**f**) Mice were transplanted as in **a–e** except mice were treated with α-PD1, α-LAG3, both, or isotype control. Total number of Tet$^{H60+}$ (**g**) and the percentage of these that were IFN-γ$^+$ (**h**) in spleen and BM. (**i**) MFI of Tim-3 and TIGIT on Tet$^{H60+}$ cells in spleen and BM. Data from (**a**) are from one experiment. Additional independent experiments comparing PD-1 blockade to isotype are in experiments analyzing LAG-3 blockade (**f**, **g**) and Tim-3 and TIGIT blockade (Supplementary Fig. 5, **b**, **h**). In sum, 22 mice received an isotype control and 17 received PD-1 blockade alone. An additional 14 mice received PD-1 blockade with another blocking agent. Experiments testing blockade of Tim-3, TIGIT and LAG3 were single experiments. For all panels, data were analyzed by an unpaired Student two-sided *t*-test. Bars are mean values, ± SEM. *$P < 0.05$; **$P < 0.01$; and ***$P < 0.001$.

uniformly induced in all T$_N$ or T$_{MH60}$-derived Tet$^{H60+}$ cells in spleen (Fig. 10a, b; Supplementary Fig. 9A) and BM (Supplementary Fig. 9B), with higher TOX MFIs in T$_{MH60}$ progeny. TOX expression correlated with PD-1 levels and Tet$^{H60}$ binding. FGK45 reduced TOX MFIs, especially in spleen, and reduced the frequency of Tet$^{H60+}$ cells with high MFIs for both TOX and PD-1 (Fig. 10b and Supplementary Fig. 9B). TOX expression in T$_N$-derived Tet$^{H60}$-negative cells was bimodal, with discrete populations of PD-1$^{high}$TOX$^{high}$ and PD-1$^{low}$TOX$^{low}$ cells (Supplementary Fig. 9C, D). FGK45 increased the frequency of Tet$^{H60}$-PD-1$^{high}$TOX$^{high}$ cells in spleen, perhaps by enhancing alloreactive CD8 cell expansion.

Tet$^{H60+}$TCF-1$^+$ cells emerged in all groups by day +7 (Fig. 10c, d; Supplementary Fig. 9E, F). These were CD39$^{low}$ with lower expression of KLRG-1, Tim-3, and PD-1 and higher expression of Ly108 (*slamf6*), relative to TCF-1$^-$ cells. This phenotype closely matches that of CD8$^+$ T cells identified as precursors for exhausted T cells in chronic LCMV infection and in tumor models[34–37,39]. CD39$^-$TCF-1$^+$ cells comprised a smaller fraction of Tet$^{H60+}$ progeny of T$_{MH60}$ than of Tet$^{H60+}$ progeny of CD8 cells from unvaccinated donors, and this frequency was not affected by FGK45, suggesting that T$_N$ may be more prone to develop into TCF-1$^+$ cells. In contrast, FGK45 reduced the fraction of T$_N$-derived Tet$^{H60+}$ cells that

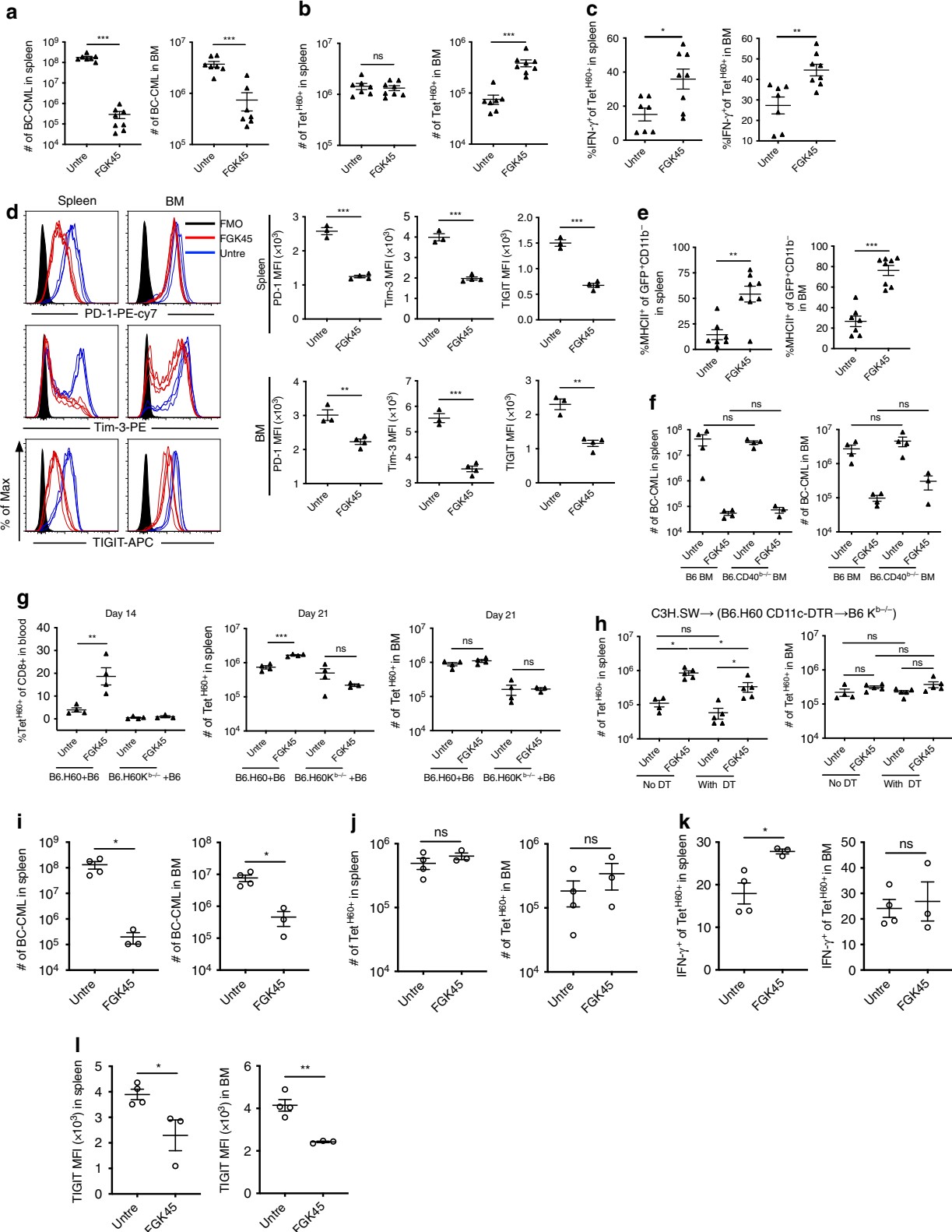

were CD39⁻TCF-1⁺ perhaps by reducing antigen exposure by creating a more effective response. In $T_N$-recipients a substantial fraction of Tet^H60-negative cells also differentiated into TCF-1⁺ CD39^low KLRG1^low TIM3^low Ly108^high PD-1^low cells (Fig. 10e, f). It is likely that these cells were alloreactive and that the generation of TCF-1⁺ cells is not restricted to those reacting to H60.

## Discussion

A barrier to reducing leukemia relapse post-alloSCT has been an incomplete understanding of the mechanisms of GVL failure. Polyclonal T cell systems and human studies have been limited by the inability to track GVL-inducing T cells, which we addressed by developing a model wherein GVL is exclusively mediated by a defined population CD8 cells reactive against H60. This

**Fig. 7 Treatment with anti-CD40 mAb on day 0 augments $T_{MH60}$-mediated GVL.** (**a–e**) B6.H60 mice were irradiated and reconstituted with B6 BM, B6. H60 BC-CML, and B6 $T_{MH60}$, with or without FGK45 administered on day 0. Mice were sacrificed on day +21 for analysis of BC-CML and Tet$^{H60+}$ cells. Total numbers of BC-CML cells and the percentages of these that are MHCII$^+$ in spleen and BM are in **a** and **e**, respectively. The total number of Tet$^{H60+}$ cells (**b**), the percentage of IFN-γ–producing Tet$^{H60+}$ cells (**c**), and the expression of PD-1, Tim-3, and TIGIT on Tet$^{H60+}$ cells (**d**) in spleen and BM are shown. For **a–c**, and **e**, data are pooled from two independent experiments with 7–8 mice per group. Panel **d** is representative of two experiments with 3–4 mice per group. (**f**) Same experimental design as in (**a**), except donor BM was from B6 or CD40$^{-/-}$ mice. Total numbers of BC-CML cells in spleen and BM are shown. Data are from one experiment (3–4 mice per group). (**g**) B6.H60 + B6→B6 and B6.H60K$^{b-/-}$ + B6→B6 BM chimeras were reirradiated and reconstituted with B6 CD40$^{-/-}$ BM and B6 $T_{MH60}$ with or without FGK45. Shown are the percentages of Tet$^{H60+}$ cells in blood at day +14, and total numbers of Tet$^{H60+}$ at day +21 in spleen and BM. Data are representative of two experiments with 3–4 mice per group. (**h**) B6.H60 CD11c.DTR$^+$→B6 K$^{b-/-}$ BM chimeras were irradiated and reconstituted with B6 BM and B6 $T_{MH60}$ with or without FGK45 on day 0 (untreated; untre). Mice received DT on day −2 and day 0 to deplete recipient DCs and were sacrificed 7 days post-transplant and numbers of Tet$^{H60+}$ in spleen and BM cells were determined. Data are representative of 2 experiments (4–5 mice per group). (**i–l**) Irradiated B6 actH60 mice were reconstituted with B6.H60 BC-CML, C3H.SW BM and $T_{MH60}$, with or without FGK45. Mice were sacrificed on day 18. Shown are number of BC-CML cells (**i**) and Tet$^{H60+}$ cells (**j**) in spleen and BM, the frequency of Tet$^{H60+}$ cells that were IFN-γ$^+$ (**k**) and the TIGIT MFI of Tet$^{H60+}$ cells (**l**). Panels **i–l** are from single repetition with 3–4 mice per group. For all panels, data were analyzed by an unpaired Student two-sided $t$-test. Bars are mean values ± SEM, $*P < 0.05$; $**P < 0.01$; and $***P < 0.001$.

permitted us to unambiguously identify and manipulate these cells to test hypotheses on the mechanisms underlying leukemia relapse. We found that GVL was limited by both ineffective antigen presentation and the development of T cell exhaustion. Importantly, we demonstrate clinically applicable strategies that enhance antigen presentation, diminish exhaustion and, critically, augment GVL. And we for the first time describe the development of TCF-1$^+$ alloreactive T cells derived from both $T_M$ and $T_N$, which could be the precursors of terminally differentiated exhausted T cells, which may fuel alloimmune responses that persist with chronic antigen.

As recipient hematopoietic cells were eliminated, effective antigen presentation declined despite a growing mass of H60-bearing BC-CML cells which, rather than stimulating T cells, contributed to their exhaustion. Antigen presentation was improved by FGK45 and DEC-H60 immunization acting through donor DCs. It was surprising that leukemia-derived antigen was so ineffectively cross-presented, even when leukemia cell death was induced by alloreactive donor CD4 cells. In contrast, caspase-9-induced leukemia apoptosis timed with T cell infusions and FGK45 improved antigen presentation and alloreactive T cell activation. In sum these data highlight strategies to enhance the efficacy of donor leukocyte infusions, which as currently applied, have limited efficacy.

However, even with vaccination or induction of leukemia apoptosis, there was less expansion of $T_{MH60}$ infused at day 14 than of $T_{MH60}$ infused at day 0. This highlights how conducive the peri-transplant period is for T cell activation, an environment that was further improved by FGK45. The principle impact of day 0 FGK45 was to increase H60-reactive T cell division driven by recipient APCs directly presenting H60. CD40 engagement on APCs upregulates MHCI, CD80 and CD86 and ligands for the TRAF-binding TNF-receptor family members 4-1BB, OX40 and GITR[40–42], which we found to be upregulated on Tet$^{H60+}$ cells from FGK45-treated mice. IL-2Rα expression was also increased on Tet$^{H60+}$ cells. In sum, these effects could have created a feed-forward process that drove expansion of Tet$^{H60+}$ cells. CD40 activation also promotes DC survival[43] which could have prolonged direct presentation of H60 by host APCs. Further delineation of the relative impacts of each of these mechanisms will be the subject of future work.

T cell exhaustion and a role for PD-1 in suppressing alloreactive T cell responses have been previously reported[12,44–50]. However, unlike most prior work, we were able to specifically track alloreactive T cells, and in GVL models, those that definitively mediate GVL. This enabled us to make additional contributions towards understanding exhaustion and how to mitigate it. Exhausted Tet$^{H60+}$ cells expressed high levels of Eomes and Blimp-1 and lost mitochondrial mass and mitochondrial membrane potential. Taken together, the

phenotypic, transcription factor and metabolic features of exhausted T cells in our experiments suggest that alloreactive T cell exhaustion in alloSCT is similar to that in chronic viral infections[17,27,31,51], which is driven by antigen exposure. Consistent with this, BC-CML increased exhaustion via direct antigen presentation and not through expression of PD-L1/L2. We also add to prior studies by demonstrating that PD-1 is the dominant inhibitory checkpoint as blockade of Tim-3, TIGIT and LAG3 did not augment GVL, though LAG3-blockade increased the fraction of Tet$^{H60+}$ cells that produced IFN-γ, consistent with studies in chronic viral infection[52].

We also for the first time demonstrate the generation of TCF-1$^+$ miHA-reactive T cell progeny of both CD8$^+$ $T_N$ and $T_{MH60}$. These cells share phenotypic properties of TCF-1$^+$ precursors of exhausted cells described in other models wherein there is also sustained antigen presentation, including being CD39$^{low}$-TIM-3$^{low}$Ly108$^+$[32,34–39]. In our experiments their frequency was reduced by FGK45 administration, suggesting that by boosting the alloresponse, FGK45 accelerated H60 clearance. It is tempting to hypothesize that these TCF-1$^+$ cells are important for GVHD maintenance, just as they sustain antiviral and antitumor responses. That alloreactive $T_{MH60}$ progeny were less likely to develop into TCF-1$^+$ cells could in part explain why $T_M$ induce less GVHD. These questions will need to be addressed in future studies.

Our studies suggest clinically applicable strategies for improving T cell immunotherapies. We targeted a hematopoietically-restricted miHA, a class of antigens which has been suggested to be ideal immunotherapy targets[5,20,21] as CD8 cells that recognize these do not cause GVHD[53]. Our results are also applicable to T cell immunotherapies that target other types of antigens that are restricted to hematopoietic or leukemia cells[54–59]. MiHA-specific T cells are most potent when given at the time of transplant, when miHAs can be directly presented by recipient APCs. Their effect is augmented by early PD-1 blockade, and more dramatically, by anti-CD40, which we found to be safe and which has been used in the clinic[60,61]. This is in contrast to safety-driven designs wherein T cells are infused remote from transplant when the major source of hematopoietic antigen is leukemia cells, which we demonstrate promote exhaustion and which are poor sources of cross-presented antigen. If anti-leukemia T cells are infused remote from transplant or when antigens are restricted to leukemia cells, their efficacy would be enhanced by antigen immunization with anti-CD40 as an adjuvant. This approach may also improve the efficacy of adoptive T cell immunotherapies against solid tumors, wherein professional APC presentation of neoantigens is likely to be very limited[62]. In patients with overt leukemia an alternative to immunization would be therapies that spare T cells but lead to rapid leukemia apoptosis to promote cross-presentation, such as gemtuzumab, an anti-CD33 immunotoxin[63].

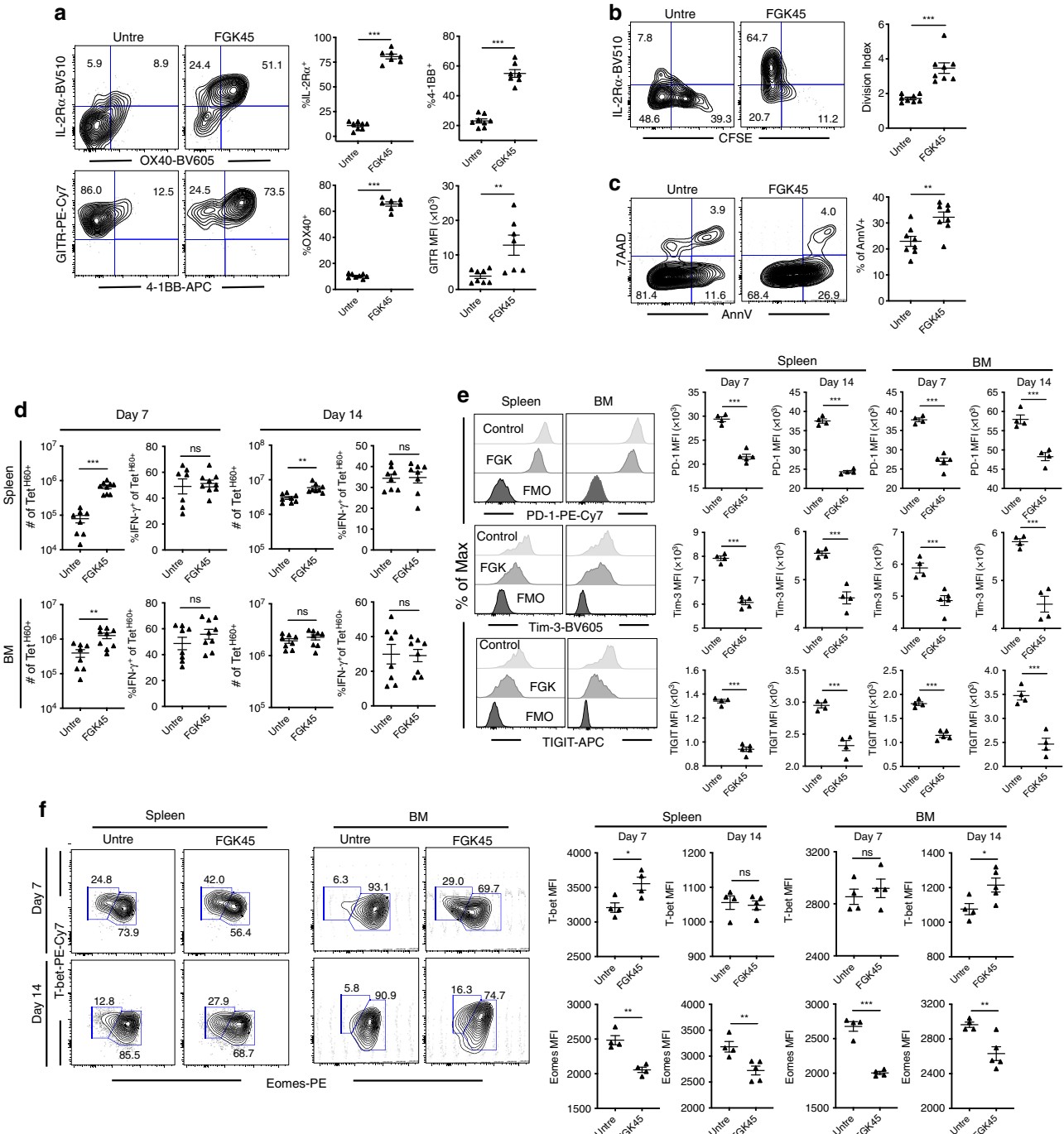

**Fig. 8 Anti-CD40 increases stimulatory and decreases inhibitory receptors on Tet^H60+ CD8 cells. a–c** B6.H60 mice were irradiated and reconstituted with B6 BM and CFSE-labeled B6 T_MH60, with or without day 0 FGK45. Cohorts were sacrificed 3 days (**a–c**) or 7 and 14 days post transplant (**d–f**) to assess proliferation, death and phenotypes of Tet^H60+ cells in spleen. FGK45 increased Tet^H60+ T cell expression of 4-1BB, IL-2Rα, OX-40 and GITR (**a**) and their division index (**b**) but increased apoptosis (**c**). Total numbers of Tet^H60+ cells and the percentages of these that are IFN-γ^+ in spleen and BM are shown in **d**. FGK45 reduced expression of PD-1, Tim-3 and TIGIT at days +7 and +14 (**e**). FGK45 also increased the fraction of cells that were Tbet^high and Eomes^low (**f**). Panels **a–d** are pooled from two independent experiments (7–9 mice per group). Panels **e**, **f** are representative of two experiments (4–5 mice per group per experiment). For all panels, data were analyzed by an unpaired Student two-sided t-test. Bars are mean values ± SEM. *P < 0.05; **P < 0.01; and ***P < 0.001.

## Methods

**Mice**. C57BL/6J (CD45.2; H-2^b), B6.SJL-Ptprca Pepcb/BoyJ (CD45.1, H-2^b), B6 CD11c-DTR[24], B6 CD40^−/−, B6 beta-2-microglobulin-deficient (β2M^−/−), B6 ubiquitin-GFP transgenic and B6 RAG^−/− and C3H.SW (H-2^b; H60^−) mice were purchased from Jackson Labs (JAX) and were bred at the University of Pittsburgh (Pitt). B6 K^b−/− mice were purchased from Taconic. OT-1 RAG1^−/− and B6-Rag2^tm1Fwa II2rg^tm1Wjl (RAG^−/−γc^−/−) mice were provided by Fadi Lakkis (Pitt).

B6.H60 mice were originally obtained from Derry Roopenian (JAX) and were bred at Pitt. B6.H60K^b−/− and B6.H60-CD11c.DTR mice were generated at Pitt. C3H. SW CD45.1^+ mice were generated by crossing C3H.SW mice to B6.CD45.1^+ mice (>10 generations). Animal breeding and experiments were performed in a specific pathogen-free animal facility in compliance with a protocol approved by Institutional Animal Care and Use Committee of the University of Pittsburgh, and we complied with all the ethical regulations.

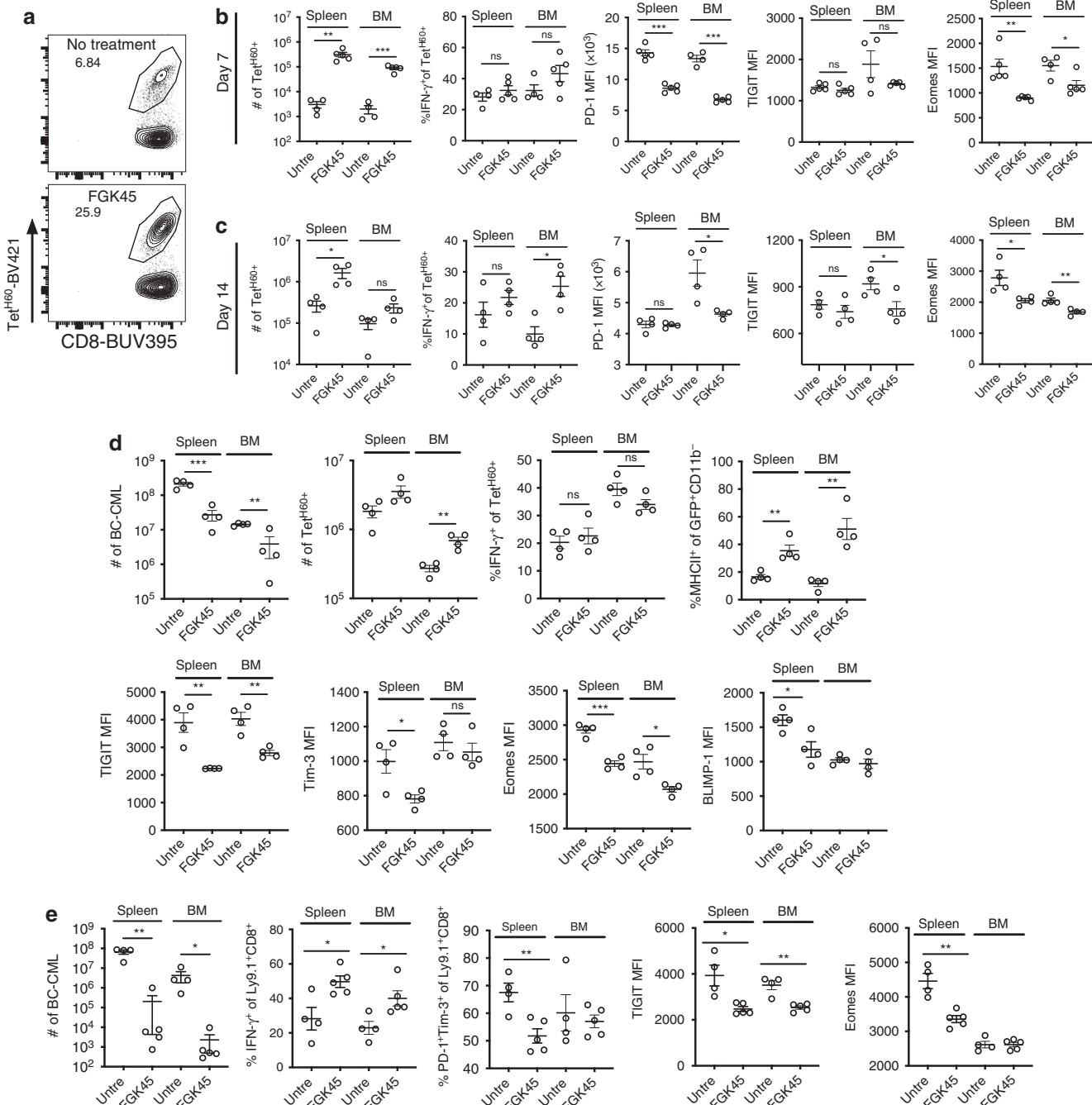

**Fig. 9 Anti-CD40 augments the anti-miHA response of naïve CD8 cells. (a–c)** Irradiated B6.H60 mice were reconstituted with C3H.SW BM and C3H.SW CD8 cells from unmanipulated donors (referred to as naïve T cells; $T_N$), with or without FGK45 administered on day 0. Representative Tet$^{H60}$ staining is in **a**. The numbers of Tet$^{H60+}$ cells, percentages that were IFN-γ$^+$, and MFIs of PD-1, TIGIT, and Eomes on days +7 and +14 are in **b**, **c**, respectively. Panels **a–c** are representative of 2 experiments ($n = 4$ or 5 per group). **d** FGK45 augments $T_N$ CD8-mediated GVL. B6.H60 mice were irradiated and reconstituted with BC-CML, B6 BM and B6 CD8 cells from unmanipulated donors, with or without FGK45. Mice were sacrificed on day +16. The numbers of BC-CML cells, Tet$^{H60+}$ cells, percentages of Tet$^{H60+}$ that were IFN-γ$^+$ and MFIs of TIGIT, Tim-3, Eomes and Blimp-1 in spleen and BM are shown. Data are from one experiment (four mice per group). **e** FGK45 augments GVL that targets other miHAs. Irradiated B6 mice were reconstituted with B6 BC-CML, C3H.SW BM and CD8 cells, with or without FGK45. Mice were sacrificed on day +18. Shown are numbers of BC-CML cells, percentages of donor Ly9.1$^+$CD8$^+$ cells that were IFN-γ$^+$ or PD-1$^+$Tim-3$^+$ and the MFIs of TIGIT and Eomes on donor CD8 cells. Data are from single experiment (4–5 mice per group). All panels were analyzed by an unpaired Student two-sided $t$-test. Bars are means ± SEM. *$P \leq 0.05$; **$P \leq 0.006$; ***$P \leq 0.007$.

**Leukemia induction**. BC-CML cells were generated as previously described[25]. Briefly, BM from 5FU-treated mice underwent two rounds of spin-infection with two MSCV2.2-based retroviruses—one encoding *bcr-abl* (coexpressing a nonsignaling human nerve growth factor receptor; NGFR) and a second expressing the NUP98/HOXA9 fusion cDNA (co-expressing GFP). iCasp9 B6.H60Kb$^{−/−}$ BC-CML was

generated by spin-infection of B6.H60K$^{b−/−}$ BM with bcr-abl and NUP98/HOXA9 retrovirus with an additional retrovirus encoding an iCasp-9 inducible suicide gene linked via a cleavable 2A-like sequence to a truncated human CD19 marker gene[26] (gift from Cliona M. Rooney; Baylor College of Medicine). Multimerization and activation of iCasp9 was achieved by in vivo treatment with AP20187 (Clontech).

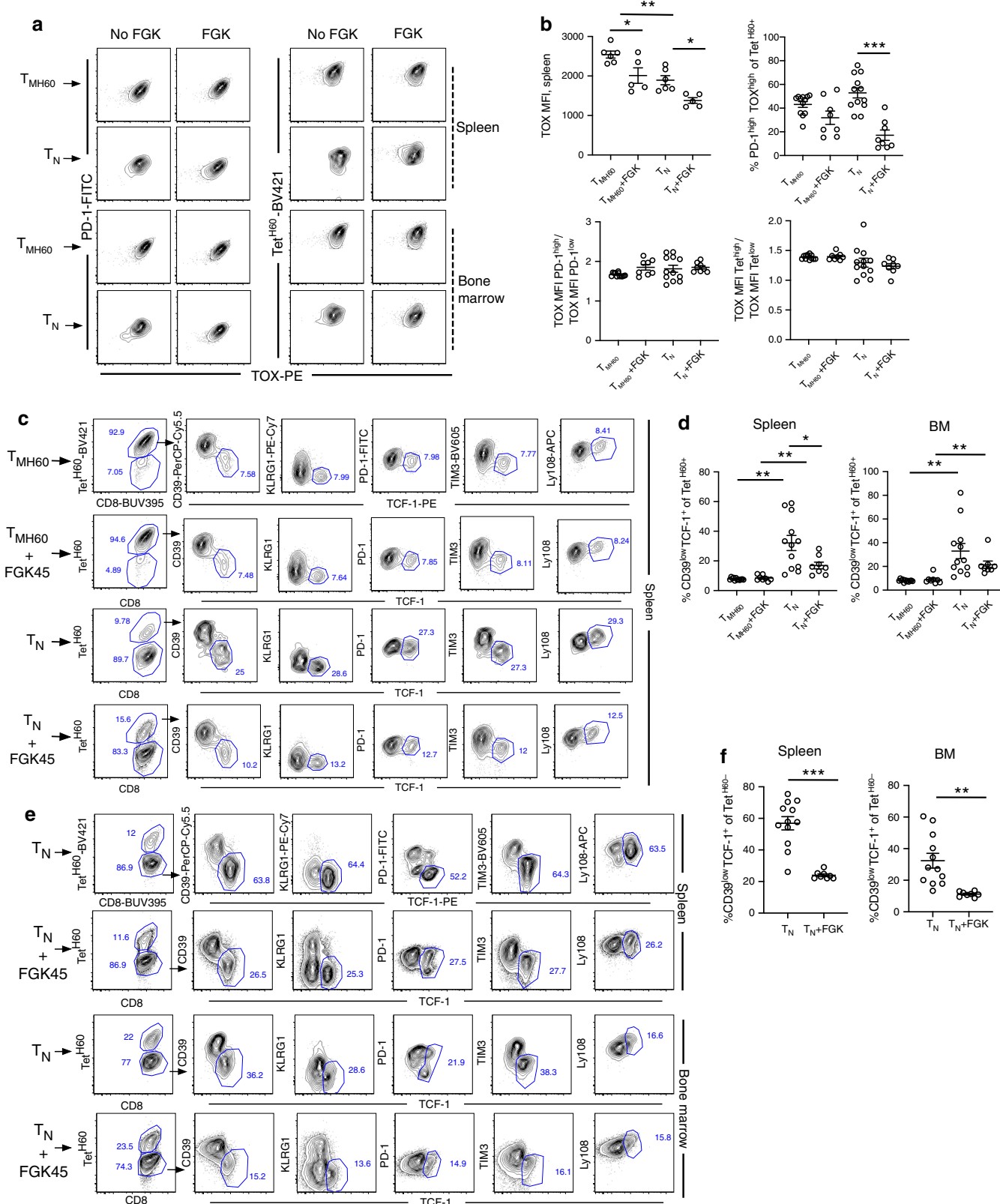

**Vaccination**. To create CD8$^+$ H60-reactive memory T cells (T$_{MH60}$) or effectors (T$_{eff}$), C3H.SW or B6 background mice were injected with 50 µg anti-DEC205-H60 (a construct encoding an antibody against DEC-205 modified to express the LTFNYRNL epitope from H60;[18] laboratory-prepared) and 50 µg of an agonist antibody against CD40 (FGK45; laboratory-prepared). T$_{MH60}$ were harvested 3 months after vaccination. To generate OT-1 T$_{eff}$, B6 mice were injected with

50 µg anti-DEC-OVA[16] (laboratory-prepared) and 50 µg FGK45 after adoptive transfer of 10$^5$ OT-I Rag1$^{-/-}$ splenocytes.

**Cell purifications**. T cell purifications from lymph node (LN) and spleen cells were performed using EasySep negative selection reagents according to the

**Fig. 10 Alloreactive TOX$^+$ and TCF-1$^+$ cells develop post-transplant.** Irradiated B6.H60 mice were reconstituted with C3H.SW BM and either C3H.SW T$_{MH60}$ (containing $10^4$ T$_{MH60}$) or $10^6$ CD8 cells from unmanipulated C3H.SW mice. Recipients were sacrificed for analysis on day +8. Representative flow cytometry (gating on CD8$^+$Tet$^{H60+}$ cells) of TOX expression versus PD-1 and Tet$^{H60}$ is in **a**. The TOX MFI was greater in Tet$^{H60+}$PD-1$^{high}$ versus Tet$^{H60+}$PD-1$^{low}$ cells (left 2 columns) and Tet$^{H60\ high}$ versus Tet$^{H60\ low}$ cells (right 2 columns). FGK45 reduced the TOX MFI and the percentage of T$_N$ Tet$^{H60+}$ progeny that were both PD-1$^{high}$ and TOX$^{high}$. These data are quantitated in **b** and in Supplementary Fig. 8. **c** Expression of TCF-1 versus CD39, KLRG1, PD-1, Tim-3, and Ly-108 of Tet$^{H60+}$ cells from spleen. **d** Frequency of Tet$^{H60+}$ cells that were CD39$^{low}$TCF-1$^+$ cells in spleen (left) and BM (right). Shown in **e** is representative staining of TCF-1 versus CD39, KLRG1, PD-1, Tim-3, and Ly-108 of Tet$^{H60}$-negative cells from spleen and BM. The percentages of Tet$^{H60-}$ cells that were CD39$^{low}$TCF-1$^+$ in spleen and BM are shown in **f**. TOX MFI in **b** are representative of two independent experiments (5–6 mice per group). Rest panels in **b**, **d** and **f** are pooled from two independent experiments ($n = 8$ in untreated group; $n = 12$ in FGK45 group). For all panels, data were analyzed by an unpaired Student two-sided *t*-test. Bars are mean values ± SEM. *$P < 0.05$; ** $P < 0.005$; ***$P < 0.0005$.

manufacturer's instructions (StemCell Technologies). Cell purities were >90% with <2% of contaminating CD4 cells. CD8$^+$ memory T cells (T$_M$) were isolated from H60-vaccinated mice by first using EasySep CD8 negative selection kit. Cells were then stained with antibodies against CD8 and CD44, followed by sorting on a FACS Aria (BD-Biosciences). Donor BM in all experiments was depleted of T cells using anti-Thy1.2 microbeads (EasySep) and is referred to as BM.

**Bone marrow transplantation**. All transplants were performed according to IACUC-approved protocols. All irradiation was from a cesium source. B6.H60 mice were irradiated with 900 cGy and reconstituted with $5 \times 10^6$ C3H.SW or B6-background BM cells, T$_{MH60}$ from C3H.SW or B6 mice, with or without BC-CML cells. T$_{MH60}$ were not sorted for Tet$^{H60+}$ cells. 200 µg of anti-NK1.1 (PKC136; lab-prepared) was given i.p. on days -2 and -1 in experiments with K$^{b-/-}$ BC-CML to diminish NK cell-mediated killing of K$^{b-}$ cells. In some experiments, fresh congenic T$_{MH60}$ were transferred to recipient mice at day +14 post BMT, following treatment with DEC-H60 (50 µg/mice), FGK45 (50 µg/mice) or both.

**BM chimeras**. To create B6.H60 CD11c.DTR→B6 K$^{b-/-}$ BM chimeras, B6 K$^{b-/-}$ mice received $2 \times 500$-cGy fractions (separated by 3 h) followed by reconstitution with $10^7$ B6.H60$^{+/-}$CD11c.DTR$^{+/-}$ BM. To create mixed BM chimeras, donor BM from B6.H60K$^{b-/-}$ or B6.H60 mice were mixed in a 1:1 ratio with B6 BM to reconstitute irradiated B6 mice. Donor BM reconstitution was verified 8 weeks after transplantation by flow cytometric analysis of peripheral blood.

**In vivo CTL assay**. To create BC-CML that had escaped a GVL response, B6.H60 mice were irradiated and reconstituted with C3H.SW BM, B6.H60 BC-CML and C3H.SW T$_M$ containing $10^4$ T$_{MH60}$. When relapsed BC-CML was harvested more than 50% of splenocytes were BC-CML cells. Fresh BC-CML cells were isolated from irradiated B6.H60 mice that were reconstituted with B6.H60 BC-CML and C3H.SW BM alone. Fresh and relapsed BC-CML cells were subjected to in vivo CTL assays as follows. To test killing directed against H60, fresh and resistant H60$^+$ BC-CML and negative control H60$^-$k$^{b-/-}$ BC-CML were injected i.v. into B6 mice that had been vaccinated 7 days prior with DEC-H60 and FGK45. To test killing directed to SIINFEKL, B6 mice were seeded with OT-1 cells followed by vaccination with DEC-OVA and FGK45. Seven days later, fresh and relapsed BC-CML cells that were or were not pulsed with SIINFEKL were injected into OT-1-seeded or unmanipulated B6 mice. Different BC-CML cells were distinguished by cell trace violet (CTV) and cell tracker DeepRed (CTR; ThermoFisher) staining. Mice were sacrificed 18 h later for BC-CML enumeration.

**Tetramer, antibodies, and other reagents**. H60 tetramers were produced by the NIH tetramer facility. Antibodies (Ab) and sources are as follows. Antibodies specific for CD44 (IM7), CD62L (MEL-14), CD45.1 (A20), CD45.2 (104), CD11b (M1/70), H2-K$^b$(AF6-88.5), I-A/I-E (M5/114.15.2), ICAM-1 (YN1/1.7.4), PD-1 (29F.1A12), Tim-3 (RMT3-23), TIGIT (1G9), 4-1BB (17B5), OX-40 (OX-86), IL-2Rα (PC61), TNF-α (MP6-XT22) KLRG1 (2F1), Ly108 (330-AJ) and Blimp-1 (5E7) were from BioLegend. GITR (DTA-1) and IFN-γ (XMG1.2) were from BD Biosciences. Eomes (Dan11mag), T-bet (4B10), TOX (TXRX10) and CD39 (24DMS1) were from ThermoFisher. CD8α (53-6.7) was laboratory prepared. LAG3 (4-10-C9) was provided by Dario Vignali and Creg Workman (Pitt).

Propidium iodide (Sigma) or Fixable Viability Dye eFluor 780 (ThermoFisher) were used to exclude dead cells. T cell restimulations were performed using the H60 peptide LTFNYRNL (Genescript) for 5 h. GolgiStop (BD Biosciences) was added for the final 3 h. Intracellular cytokine staining was performed using the BD Cytofix/Cytoperm kit. Transcription factor and Annexin V staining was performed using the FoxP3 staining (ThermoFisher) and Annexin V (BioLegend) kits, respectively. For metabolism assays cells were pulsed with 50 mM 2-NBDG (ThermoFisher) in FBS-free media for 30 min at 37 °C. Cells were surface stained and loaded with MitoTracker DeepRed or TMRE (ThermoFisher). For antibody blocking experiments, BC-CML-bearing mice were injected with 200 µg/mouse αPD-1 (RMP1-14, BioLegend), 200 µg/mice αLAG3 (C9B7W;[64] gift of Dario Vignali and Creg Workman), 200 µg/mice αTim-3 (RMT3-23, BioLegend), 500 µg/mice αTIGIT (10A7, Genentech) or

respective murine or rat isotype controls (BioXCell), i.p. three times per week for 2 weeks starting from day 7 post BMT. Diphtheria toxin (DT) was from List Biological Laboratories. Flow cytometry was performed on a BD LSR II (BD) and data was analyzed using FlowJo (v10.5.3).

**Statistical analysis**. Bars on scatter plots are mean values. Statistical significance was calculated using an unpaired Student two-sided *t*-test (GraphPad Prism) except in CTL assays wherein a paired two-sided *t*-test was employed to compare fresh and relapsed BC-CML cells in the same mice.

**Reporting summary**. Further information on research design is available in the Nature Research Reporting Summary linked to this article.

## Data availability
The authors declare that all of the data supporting the findings of this study are available within the paper and its supplementary information files. Raw data are included in the Source data file. Source data are provided with this paper.

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

## Acknowledgements

This work was supported by R01 HL117855 with additional support from the UPMC Hillman Cancer Center and the UPMC Immune Transplant and Therapy Center. We also thank the animal technicians at the University of Pittsburgh for their diligent care.

## Author contributions

W.D.S. conceived of experiments, analyzed data and wrote the paper. M.Z. conceived and executed experiments, analyzed data and wrote the paper. F.S. provided technical advice and performed experiments. K.Z. and S.R. assisted in performing experiments.

## Competing interests

W.D.S. declares that he is a founder, shareholder and consultant for BlueSphere Bio. The other authors declare no competing interests.
