## [Peer Review File · Nature Communications]

Reviewers' comments:

Reviewer #1 (Remarks to the Author):

Zhou et al report on graft-versus-leukemia studies in a model system that relies on T cells that recognize the minor histocompatibility antigen H60. The authors show that Leukemia-derived H60 was inefficiently cross-presented and they use different approaches to enhance GVL effects including an agonistic CD40 antibody, blockade of PD1, Tim-3, TIGIT or LAG3. T cell exhaustion and leukemia burden are used as major read-outs. The main finding is that FGK45 administered at the time of transplant was that the agonistic CD40 antibody was most potent at enhancing GVL effects.

Major comments:

1. H60 is an artificial antigen, in patients multiple hematopoietic antigens (including major MHC molecules such as HLA-DP1 which are not matched) are recognized by the donor T cells after allo-HSCT. The authors should reproduce their findings in a different leukemia model that relies at least on one additional antigen. This would make their finding relevant for the clinic.
2. FGK45 alone augmented the anti-H60 response, even in the absence of BC-CML, - this shows that there were sources of H60 other than the leukemia derived H60. How can the authors be sure that the TetH60+ (H60 reactive) T cells are then really leukemia reactive?
3. The authors state that the anti CD40 antibody was most effective - why did they not perform a survival study of the leukemia bearing mice?
4. The authors use a CML model (Figure 1) - CML is hardly anymore an indication for allo-HSCT. The studies should focus on AML.
5. Figure 6: anti PD1 treatment after allo-HSCT causes GVHD -did the authors analyze GVHD target organs?
6. The H60-expressing mouse that relies on bone marrow (BM) transduced with retroviruses that express cDNAs encoding the human bcr-abl and NUP98/HOXA9 translocations is artificial. Genetic models of AML are more relevant for the clinic.
7. The authors show that a-CD40 at time of transplantation augments GVL -this is very early treatment - in the clinic relapse is treated when it shows MRD positivity or hematological relapse. Such scenario should be tested by the authors (established AML treated with a-CD40).

Reviewer #2 (Remarks to the Author):

In this manuscript, Zhou et al. examined mechanisms of the failure in graft-versus-leukemia (GVL) using a mouse model wherein GVL is mediated by minor histocompatibility antigen H60-specific T cells that are trackable with MHC tetramer. In this model, H60 specific memory CD8 T cells generated after vaccination were adoptively transferred into irradiated recipient mice along with T cell-depleted bone marrow as well as H60 expressing leukemia cells. They found that H60-specific memory CD8 T cells failed to control the expansion of the leukemia cell population due to inefficient H60 antigen presentation. Instead, H60-antigen presented by leukemia cells contributed to the differentiation of these memory CD8 T cells into exhausted T cells. Such exhausted CD8 T cells included a PD-1+ Tcf1+ subset, which was previously observed during chronic viral infection and in solid tumors. PD-1 blockade diminished T cell exhaustion and improved GVL. Furthermore, the authors showed that treatment with anti-CD40 antibody augmented H60-specific memory CD8 T cell-mediated GVL. Similar to what was seen in the model of memory CD8 T cell transfer, naïve CD8 T cells differentiated into exhausted T cells after adoptive transfer.

Overall, this work provides a comprehensive analysis of how CD8 T cells respond to leukemia cells and also shows the mechanisms of the failure in GVL as well as a potential therapeutic approach to improve GVL. This manuscript could be improved if the authors address the following issues and questions.

1. In the mouse model used in this study, the authors irradiated H60-expressing B6 mice and then donor bone marrow, H60-specific memory CD8 T cells, and H60+ blast crisis chronic myelogenous leukemia (BC-CML) were adoptively transferred into these irradiated mice. Why were H60-expressing B6 mice used as a recipient instead of H60 negative regular B6 mice? This information will be helpful for readers to understand the experiments in this manuscript.

2. In Fig. 1C, D, and E, a burden of BC-CML is shown in the presence of H60-specific CD8 T cells. Do these H60-specific CD8 T cells contribute to the control of BC-CML? Does the expansion of the BC-CML population further increase if H60-specific CD8 T cells are not transferred?

3. Experimental designs of supplemental Fig. 2D and E are not clear. In these experiments, bone marrow cells were transplanted into irradiated B6.H60 mice but the designs give the impression that bone marrow cells were isolated from B6.H60 mice. Please revise them.

4. On pages 12 and 20, the authors mentioned that they were unable to detect H60-specific CD8 T cells that were PD1+ CXCR5+, a subset shown to respond to PD-1-blockade in LCMV infection. What clone and fluorochrome were used for staining of CXCR5 (there is no information in the MM section)? CXCR5 expression in CD8 T cells is dim compared with B cells, and specific clone(s) as well as brightest fluorochrome are required to detect CXCR5 on exhausted CD8 T cells. Alternatively, the existence of CXCR5 can be transcriptionally confirmed. Do the authors have the data of CXCR5 mRNA levels in PD1 positive Tim3 low H60-specific CD8 T cells (compared with those in PD1 positive Tim3 high H60-specific CD8 T cells or naïve CD8 T cells)? Since expression patterns of other markers (Tcf-1, CD39, KLRG1, and Ly108) are very similar to the LCMV system, the Reviewer is wondering if CXCR5 staining would fail due to a technical issue. If the authors do not have conclusive data about CXCR5 expression, the Reviewer suggests removing these parts from the manuscript.

5. It is very difficult to distinguish the lines in the histograms in Fig. 7D. Please revise them.

6. On page 13 line 11, the authors mentioned: "FGK45 primarily acted on host cells as there was no reduction in GVL in recipients of CD40^{-/-} BM (Fig. 7F)." What are the host cells? Do host cells mean recipient cells? Please clarify them.

7. It is very difficult to understand the rationale, the experimental design, and the data of Fig. 7G. Why were mixed bone marrow chimera mice made? Why were these mixed bone marrow chimera mice re-irradiated for transplantation? This part needs more information for a better understanding of the data.

8. On page 16 line 15, (Tn) should be right after "CD8 T cells from either unmanipulated C3H.SW donors".

9. In Fig. 10A and B, it will be better to show Tox expression levels in unstimulated naïve CD8 T cells as a negative control.

10. On page 16 the last line to page 17 the first line, the authors said "TOX expression in TetH60-negative cells in Tn-recipients was bimodal, with discrete populations of PD-1 High TOX High and PD-1 low TOX low cells (Fig. 10E). FGK45 increased the frequency of TetH60 negative PD-1 high TOX high cells in spleen, perhaps by enhancing alloreactive CD8 cell expansion." However, Fig. 10E analyzes Tcf-1 expression (no TOX expression data). So, the Reviewer thinks that Tcf1 expression in TetH60-negative cells in Tn-recipients was bimodal. Also, Fig. 10E shows that PD-1 high cells are Tcf-1 low. Conversely, PD-1 low cells are Tcf-1 high. Please revise and clarify this part.

Reviewer #1 (Remarks to the Author):

Zhou et al report on graft-versus-leukemia studies in a model system that relies on T cells that recognize the minor histocompatibility antigen H60. The authors show that Leukemia-derived H60 was inefficiently cross-presented and they use different approaches to enhance GVL effects including an agonistic CD40 antibody, blockade of PD1, Tim-3, TIGIT or LAG3. T cell exhaustion and leukemia burden are used as major read-outs. The main finding is that FGK45 administered at the time of transplant was that the agonistic CD40 antibody was most potent at enhancing GVL effects.

While the efficacy of FGK45 was a major novel finding, we believe our manuscript makes other important observations regarding the mechanisms of leukemia relapse and T cell exhaustion, including some listed by Reviewer 1. Our major conclusions include: 1) The availability of productively presented miHA becomes a limiting factor in alloreactive T cell stimulation; 2) This limitation of antigen can be addressed by both vaccination with an antibody against DEC205 which was modified to express the targeted miHA (DEC-H60) and by anti-CD40 (FGK45), alone or in combination with the DEC-H60; 3) FGK45 given to promote antigen presentation 14 days post-transplant works through activating donor CD11c⁺ dendritic cells; 4) Leukemia cells are not a good source of cross-presented antigen unless they are induced to undergo apoptosis, which we accomplished with an iCasp9-expressing, K^{b/-} leukemia; 5) Leukemia cells do not effectively stimulate alloreactive T cells in vivo; 6) MiHA-reactive T cells become exhausted and this exhaustion is intensified by the presence of leukemia cells due to direct antigen presentation by the leukemia cells and not due to leukemia expression of PD-ligands; 7) The metabolic and transcription factor profile of exhausted miHA-reactive T cells parallels what is seen in chronic viral infection more so than what is seen in the tumor microenvironment; 8) PD-1 inhibition, but not inhibition of TIGIT, TIM-3 or LAG-3 (alone or in combination with PD-1 blockade), diminishes exhaustion and promotes GVL; 9) FGK45 given at the time of transplant dramatically improves GVL; 10) FGK45 at the time of transplant acts by promoting direct miHA presentation by recipient hematopoietic antigen presenting cells, but recipient dendritic cells are not an essential antigen presenting cell subset; 11) FGK45 promoted GVL with ubiquitous host H60 expression (in B6actH60 transgenic recipient mice); 12) FGK45 promoted GVL in the C3H.SW → B6 model wherein miHAs other than H60 were targeted; 13) FGK45 improved the anti-H60 response and GVL mediated by naïve CD8 cells; and 14) For the first time we report the development of TCF-1⁺ miHA-reactive T cells which share the phenotype of T cells that sustain antiviral responses in situations of chronic viral infection. These conclusions (and others) are supported by 94 panels of data.

Major comments:

1. H60 is an artificial antigen, in patients multiple hematopoietic antigens (including major MHC molecules such as HLA-DP1 which are not matched) are recognized by the donor T cells after allo-HSCT. The authors should reproduce their findings in a different leukemia model that relies at least on one additional antigen. This would make their finding relevant for the clinic.

H60 is not an artificial antigen but is a naturally occurring miHA that was molecularly identified as a target of CD8 responses by B6 CD8 cells against BALB.B cells [1-5]. Other miHAs were identified at the genetic level by backcrossing chromosome segments from BALB.B to B6 mice and H60 was encoded by one of these segments [6-10]. B6.H60 mice are congenic for H60—that is the gene crossed from BALB.B to B6 by Derry Roopenian. Other mouse strains express H60, including 129 mice, and when 129 mice are transplanted with a mix of B6 CD4 and CD8 cells, Tet^{H60+} CD8 cells develop and are present in spleen lymph nodes, bone marrow and GVHD target tissues (Figure 1). We also want to highlight the fact that T cell responses against miHAs demonstrate “immunodominance”. That is, even when multiple target antigens are available, the immune response tends to focus on one or a few. To our knowledge, immunodominance was first described in responses to miHAs [1, 5, 11-13], but it has also been observed in T cell responses to pathogens. Importantly, H60 is naturally immunodominant in the alloBMT setting [5].

In 8/8 HLA-matched unrelated donor transplants some “nonpermissive” mismatches at HLA-DP increase the risk for GVHD. However, GVHD develops in DP-matched transplants including HLA-matched sibling transplants. While some transplants are mismatched at HLA molecules, in 2019 according to the CIBMTR

Figure 1. Tet^{H60} CD8 cells develop in the B6 (H-2^b) → 129 (H-2^b) GVHD model. Irradiated 129 mice were reconstituted with B6 BM and 10⁶ CD8 and 10⁶ CD4 cells. Mice were sacrificed on day +28 for analysis. Plots are gated on live CD8⁺ cells. BM, bone marrow; IEL, intraepithelial lymphocytes; LP, lamina propria.

in the US about 25% and 40% of transplants were with HLA-matched sibling donors and unrelated donors, respectively. Based on these numbers we believe our use of a transplant model that does not have a mismatch at an MHC molecule is relevant to clinical practice.

As was explained in detail in the manuscript, it was necessary to develop a system wherein all GVL-inducing T cells could be tracked and accounted for. This was the rationale for and the advantage of the H60 system. We also point out that the development of T cell therapies that target hematopoietically-restricted miHAs or leukemia-specific antigens has been and remains a goal of many research groups [14-33] and a clinical trial of a T cell product transduced with a TCR against the hematopoietically-restricted miHA HA1 is underway (NCT03326921).

There really is no way we can develop an entire second model antigen system. There has not been an intense effort to molecularly identify mouse miHAs and therefore there are not suitable reagents available for a second system using natural minor histocompatibility antigens. The alternative would be the use of ovalbumin (OVA) transgenic mice. But these mice express ovalbumin ubiquitously and are not a good model for a relatively hematopoietically-restricted antigen. Even if we were to undertake this using OVA as model antigen we would need to cross these mice to K^{b-/-} mice and make new panels of leukemias (K^{b-/-}, PD-L1/L2^{-/-}, iCasp9-expressing), vaccinate new mice to make memory cells and use bone marrow chimeric mice as transplant recipients so as to make OVA hematopoietically restricted. It would be years of work, be very expensive, and I don't think we would have fundamentally different results as in the H60 model which we document in great detail. We hope Reviewer 1 is sympathetic to these very practical issues.

As Reviewer 1 knows, systems in which T cell responses can be tracked and in which at least the dominant, if not only response, is directed to a single antigen have been pivotal in elucidating fundamental aspects of T cell immunity. This powerful approach has been applied to mouse systems studying responses to pathogens, tumors, transplanted organs, GVHD, asthma and autoimmunity, as well as to models of self-tolerance [34-80]. Frequently T cell receptor (TCR) transgenic T cells that recognize the targeted antigen are used in such experimental systems, including those against OVA (OT-I CD8 and OT-II and DO11.10 CD4) [57, 58, 81] and LCMV-derived GP33 and GP66 (P14 CD8 and SMARTA CD4) [37-41, 43-47, 82-87]. These models were pivotal in the discovery of T cell exhaustion and in the deployment of checkpoint-blockade in the treatment of cancer and manuscripts employing such models have been widely published in high impact journals. A goal of our studies was to employ a similar "tool kit" to understand mechanisms of GVL-resistance.

Finally, we remind Reviewer 1 that our manuscript has substantial data in models wherein H60 is either not the only target miHA or is not a miHA at all.

1. We performed experiments in the C3H.SW → B6.H60 model using CD8⁺ T cells from *unmanipulated* C3H.SW donors wherein T cells responded to multiple miHAs including H60. In this system, Tet^{H60} CD8 cells had evidence of exhaustion. FGK45 improved the anti-H60 response as it did in the T_{MH60} systems. In new data we have included in the revised manuscript, we show evidence of exhaustion in a subset of the Tet^{H60} CD8 cells. Their total number was increased by FGK45 by a similar magnitude as were Tet^{H60} CD8 cells, indicating that FGK45 acted on T cells responding to miHAs other than H60.
2. FGK45 improved GVL in the C3H.SW → B6 model (not B6.H60 hosts) wherein H60 **was not** a miHA and this was certainly due to improving the T cell responses against miHAs. In these experiments the

- fraction of IFN- γ ⁺ CD8 cells was increased by FGK45, consistent with a reduction in T cell exhaustion.
- We also demonstrated the development of Tet^{H60+} and Tet^{H60-} TCF-1⁺ and TOX⁺ progeny of naïve donor CD8 cells in the C3H.SW → B6.H60 system.

2. FGK45 alone augmented the anti-H60 response, even in the absence of BC-CML, - this shows that there were sources of H60 other than the leukemia derived H60. How can the authors be sure that the TetH60+ (H60 reactive) T cells are then really leukemia reactive?

It is well established that host hematopoietic cells are key sources of miHAs and host antigen presenting cells that directly present miHAs are particularly important [17, 88]. Therefore, it is expected that transplanting B6.H60 or actH60 mice without leukemia would yield strong anti-H60 responses, with or without FGK45. This is also the clinical situation. When most patients (aside from those with chronic phase CML) are transplanted, their leukemia is in remission or at least present at a low level and therefore the major source of miHAs is not leukemia cells, but other nonmalignant host cells [89, 90]. GVHD also develops when alloSCT is employed in treatment of nonmalignant disorders such as sickle cell disease and aplastic anemia. Further support for this can be found for the importance of nonmalignant host miHAs in the present work wherein there was no anti-H60 response in B6 (H60-negative) mice transplanted with H60⁺ BC-CML cells, be they wild type or K^b-.

We know that H60-reactive T cells are leukemia-reactive because transplanted T_{MH60} have no impact on H60-negative leukemias but mediate potent GVL against H60⁺ leukemias. This is published [91] but we failed to specifically call this out in the manuscript and we apologize for this omission. We have modified the manuscript to make this point explicitly. For the benefit of Reviewer 1 we have included the key figure from that paper in the rebuttal (Figure 2).

Figure 2. T_{MH60}-mediated GVL requires that the target leukemia expresses H60. B6.H60 mice were irradiated and reconstituted with T cell depleted C3H.SW BM, B6.H60 or B6 (H60⁻) mBC-CML cells, with no T cells or with CD8⁺ T_M from DEC-H60-vaccinated or unmanipulated C3H.SW donors or CD8⁺ T_N from unmanipulated C3H.SW mice. (A) Survival. All deaths were from leukemia. $P < 0.0001$ comparing T_{MH60} recipients of B6.H60 vs B6 mBC-CML. (B) Numbers of NGFR⁺EGFP⁺ cells in peripheral blood at day 14. $P < 0.0001$ comparing T_{MH60} recipients of B6.H60 versus B6 H60⁻ mBC-CML cells. $P = 0.5$ comparing B6 mBC-CML recipients of T_{MH60} or T_M cells.

- The authors state that the anti CD40 antibody was most effective - why did they not perform a survival study of the leukemia bearing mice?

Respectfully, we do not believe that survival is a necessary endpoint in these types of studies. By sacrificing mice, we can enumerate and further characterize both leukemia cells and T cells. This yields a quantitative measure of an effect, rather than just a binary alive/dead readout. We don't think there is any doubt that 3-log10 and nearly 10-fold reduction in BC-CML cells in the spleen and BM (respectively) in FGK45-treated mice on day +21 (see Figure 7A) would not have improved survival.

- The authors use a CML model (Figure 1) - CML is hardly anymore an indication for allo-HSCT. The studies should focus on AML.

The model we use is a *myeloblastic* leukemia (BC-CML=blast crisis CML) and not chronic phase CML. It is still recommended that patients with blast crisis CML undergo alloSCT. Moreover, the biology of myeloblastic CML has similarities to de novo AML. The BC-CML cells we employed have both a dysregulated tyrosine kinase (bcr-abl) that drives proliferation and a dysregulated transcription factor that inhibits differentiation [92]. This two hit model of AML was a hypothesis of Gary Gilliland [93, 94], who first created the bcr-abl/NUP98-HOXA9 leukemia and the general principles of it have been broadly confirmed by subsequent genome sequencing of many subtypes of AML [95]. We also point out that upregulation of HOXA9 is a common endpoint of other molecular changes that drive leukemia pathogenesis [96], affirming the relevance of our BC-CML to de novo AML. To our knowledge, our lab was the first to use models of chronic phase CML (bcr-abl as the lone oncogene) and BC-CML in GVL experiments and the BC-CML model is increasingly being used due to its biological relevance to human AML [64, 97-102] relative to cell lines with little relevance to human leukemias and of unknown molecular pathogenesis (C1498, EL4, A20, P815) used in many prior GVL papers.

5. Figure 6: anti PD1 treatment after allo-HSCT causes GVHD -did the authors analyze GVHD target organs?

We did not. We've previously published that targeting H60 in B6.H60 mice does not cause GVHD even with a very large anti-H60 response [91] due to the hematopoietic restriction of H60. We agree that GVHD is a risk of PD-1 blockade in this setting.

6. The H60-expressing mouse that relies on bone marrow (BM) transduced with retroviruses that express cDNAs encoding the human bcr-abl and NUP98/HOXA9 translocations is artificial. Genetic models of AML are more relevant for the clinic.

As discussed above, the leukemia model we use *is* a genetic model (induced by human oncogenes) and is biologically relevant to AML in general. We believe the BC-CML model to be superior to the many cell lines that have been used in GVL papers published in high profile journals such as the cell lines EL4, A20, C1498, MBL-2, MMB3.19, BCL1 and P815. We also feel BC-CML compares favorably to solid tumor lines used in many papers. B16 melanoma-derived (including those that express OVA and GP33), MC38 colon cancer cells, TRAMP (induced by SV40 T antigen which has no role in human prostate cancer), and many others have been established as cell lines that grow in culture which is not physiologic (parenthetically, BC-CML cells cannot be propagated in culture but only mice). Yet, these are used in high impact publications. We have used the MLL-AF9 retrovirus-induced leukemia model [103] as have others. But even for this leukemia, there is a long latency after retroviral transduction due to requirements for presumed second hits, which are not defined, whereas bcr-abl and NUP98-HOXA9 are necessary and sufficient for BC-CML generation.

Also, it would be quite difficult to create the various gene modified leukemias we have used without the ability to use retroviral transduction of gene deficient bone marrow. More recent transgenic/congenic AML models are frequently on complex backgrounds (compound knockin, cre expressing) and it would be difficult to cross these to the B6.H60 background. Our manuscript employed numerous leukemias including those that are gene-modified (H60⁺, H60⁻, H60 K^{b/-}, K^{b/-/-}, PD-L1/L2^{-/-}, iCasp9, K^{b/-} iCasp9) and it would have been onerous to create these from these mice or create a second set of leukemias expressing a second target antigen. We actually have been working on using CRISPR-Cas9 approaches to edit such leukemias, and so far, it has not been easy. We hope that Reviewer 1 is sympathetic to these issues.

7. The authors show that a-CD40 at time of transplantation augments GVL -this is very early treatment - in the clinic relapse is treated when it shows MRD positivity or hematological relapse. Such scenario should be tested by the authors (established AML treated with a-CD40).

Respectfully, we disagree. Our goal was to understand the mechanisms of relapse and based on that knowledge, to test approaches to decrease it. All patients who relapse must have viable leukemia after the conditioning regimen and therefore also at the time of donor cell infusion. The approach of irradiating mice followed by the injection of donor BM, T cells and a dose of leukemia cells is the longstanding and really ubiquitous approach for modeling this. I am sure that this design is employed in nearly all mouse GVL studies.

We envision anti-CD40 to be used at the time of transplant as was tested in our experiments and not as monotherapy to treat relapse. Finally, we did show that anti-CD40 combined with donor T_{MH60} improves GVL when given at day 14 post-transplant when mice had a large leukemia burden (Figure 3D).

Reviewer #2 (Remarks to the Author):

In this manuscript, Zhou et al. examined mechanisms of the failure in graft-versus-leukemia (GVL) using a mouse model wherein GVL is mediated by minor histocompatibility antigen H60-specific T cells that are trackable with MHC tetramer. In this model, H60 specific memory CD8 T cells generated after vaccination were adoptively transferred into irradiated recipient mice along with T cell-depleted bone marrow as well as H60 expressing leukemia cells. They found that H60-specific memory CD8 T cells failed to control the expansion of the leukemia cell population due to inefficient H60 antigen presentation. Instead, H60-antigen presented by leukemia cells contributed to the differentiation of these memory CD8 T cells into exhausted T cells. Such exhausted CD8 T cells included a PD-1+ Tcf1+ subset, which was previously observed during chronic viral infection and in solid tumors. PD-1 blockade diminished T cell exhaustion and improved GVL. Furthermore, the authors showed that treatment with anti-CD40 antibody augmented H60-specific memory CD8 T cell-mediated GVL. Similar to what was seen in the model of memory CD8 T cell transfer, naïve CD8 T cells differentiated into exhausted T cells after adoptive transfer.

Overall, this work provides a comprehensive analysis of how CD8 T cells respond to leukemia cells and also shows the mechanisms of the failure in GVL as well as a potential therapeutic approach to improve GVL.

We appreciate this very positive assessment and thorough reading of our manuscript.

This manuscript could be improved if the authors address the following issues and questions.

1. In the mouse model used in this study, the authors irradiated H60-expressing B6 mice and then donor bone marrow, H60-specific memory CD8 T cells, and H60+ blast crisis chronic myelogenous leukemia (BC-CML) were adoptively transferred into these irradiated mice. Why were H60-expressing B6 mice used as a recipient instead of H60 negative regular B6 mice? This information will be helpful for readers to understand the experiments in this manuscript.

In alloSCT alloreactive GVL-inducing T cells target recipient (patient) miHAs which are expressed in varied recipient tissues, including hematopoietic cells. Many, if not most miHAs that are expressed in hematopoietic cells are also expressed in leukemia cells. It has long been a goal in the field to target miHAs that are relatively restricted to hematopoietic cells as when such miHAs are targeted by CD8 cells, no GVHD is induced. So, our model (wherein recipients express the miHA H60) accurately recapitulates the clinical situation and also models the targeting of a miHA that is relatively restricted to the hematopoietic system. One of our questions was how well leukemia-derived H60 could support alloreactive CD8 cell expansion. This was to model the situation wherein T cells are given remote from transplant (after hematopoietic antigen is largely gone) or when leukemia-specific neoantigens were targeted. We found that leukemia-derived H60 (when the host otherwise lacked H60) did not drive T cell expansion (See manuscript Figure 4).

2. In Fig. 1C, D, and E, a burden of BC-CML is shown in the presence of H60-specific CD8 T cells. Do these H60-specific CD8 T cells contribute to the control of BC-CML? Does the expansion of the BC-CML population further increase if H60-specific CD8 T cells are not transferred?

We have published that anti-H60 CD8 cells can mediate potent GVL. Importantly, they mediate no GVL against H60-negative BC-CML [91]. We now realize that we neglected to make this key point in the manuscript, and we have corrected this omission in the revision. We have included the key figure from the paper that describes this result in our rebuttal (Figure 2). In brief, irradiated B6.H60 mice were reconstituted with C3H.SW BM, with or without C3H.SW T_{MH60}. One cohort received B6.H60 BC-CML and one cohort received B6 H60-negative BC-CML. As shown, T_{MH60} mediated no GVL (as measured by survival or leukemia cell numbers) against B6 H60-negative BC-CML.

3. Experimental designs of supplemental Fig. 2D and E are not clear. In these experiments, bone marrow cells were transplanted into irradiated B6.H60 mice but the designs give the impression that bone marrow cells were isolated from B6.H60 mice. Please revise them.

We are sorry for any confusion here. The labels over the diagonal arrows were meant to show the identity of the donor BM. We have revised the figure so as to make the designs more clear. We thank Reviewer 2 for this suggestion.

4. On pages 12 and 20, the authors mentioned that they were unable to detect H60-specific CD8 T cells that were PD1+ CXCR5+, a subset shown to respond to PD-1-blockade in LCMV infection. What clone and fluorochrome were used for staining of CXCR5 (there is no information in the MM section)? CXCR5 expression in CD8 T cells is dim compared with B cells, and specific clone(s) as well as brightest fluorochrome are required to detect CXCR5 on exhausted CD8 T cells. Alternatively, the existence of CXCR5 can be transcriptionally confirmed. Do the authors have the data of CXCR5 mRNA levels in PD1 positive Tim3 low H60-specific CD8 T cells (compared with those in PD1 positive Tim3 high H60-specific CD8 T cells or naïve CD8 T cells)? Since expression patterns of other markers (Tcf-1, CD39, KLRG1, and Ly108) are very similar to the LCMV system, the Reviewer is wondering if CXCR5 staining would fail due to a technical issue. If the authors do not have conclusive data about CXCR5 expression, the Reviewer suggests removing these parts from the manuscript.

We did see some CXCR5 staining in the Tet^{H60}- population. However, we agree with Reviewer 1 that we did not examine this question as fully as other aspects of GVL-resistance and we have removed all references to this work from the manuscript.

5. It is very difficult to distinguish the lines in the histograms in Fig. 7D. Please revise them.

We apologize for how difficult this panel was to visualize and we have revised it to be in color. We thank Reviewer 2 for this suggestion.

6. On page 13 line 11, the authors mentioned: "FGK45 primarily acted on host cells as there was no reduction in GVL in recipients of CD40^{-/-} BM (Fig. 7F)." What are the host cells? Do host cells mean recipient cells? Please clarify them.

"Host cells" are the cells of the mouse that was being transplanted, as is used in "graft-versus-host disease". But we understand how the usage of "host" outside of this context would be confusing. We have now defined "host" in the manuscript.

7. It is very difficult to understand the rationale, the experimental design, and the data of Fig. 7G. Why were mixed bone marrow chimera mice made? Why were these mixed bone marrow chimera mice re-irradiated for transplantation? This part needs more information for a better understanding of the data.

We are sorry for any confusion. The goal here was to create recipient mice that had hematopoietic H60 that could either be directly presented (must have both K^b and H60 on the same cell) or in which H60 was available but could only be cross-presented (one cell is K^{b-/-} H60⁺ and the other cell in the mixed chimeras are K^{b+/+} but H60-negative). We created these mice by making mixed BM chimeras which we then reirradiated to study the anti-H60 T cell response. We wrote in the manuscript, "FGK45 only augmented the Tet^{H60+} response in the B6.H60+B6→ B6 chimeras in which H60 was directly presented whereas there was no effect in B6.H60 K^{b-/-}+B6→ B6 chimeras wherein FGK45 could only promote H60 cross-presentation by B6 K^b-intact APCs (Fig. 7G)." We have expanded this explanation as follows to improve clarity:

FGK45 primarily acted on recipient cells as there was no reduction in GVL in recipients of CD40^{-/-} BM (Fig. 7F). It was possible that FGK45 increased direct presentation of K^b:LTFNYRNL; alternatively or in addition it could have improved cross-presentation. To address this, we made mixed BM chimeras

in which recipient APCs could directly present H60 ([B6.H60+B6]→B6 chimeras) or only cross-present H60 ([B6.H60 K^{b-/-}+B6]→B6 BM chimeras). After 8 weeks, these chimeras were reirradiated and transplanted with B6 CD40^{-/-} BM and B6 T_{MH60}, with or without FGK45 (design, Supplemental Fig. 6A). FGK45 only augmented the Tet^{H60+} response in the (B6.H60+B6)→B6 chimeras in which H60 was directly presented whereas there was no effect in (B6.H60 K^{b-/-}+B6)→B6 chimeras wherein FGK45 could only promote H60 cross-presentation by B6 K^b-intact APCs (Fig. 7G).

8. On page 16 line 15, (Tn) should be right after “CD8 T cells from either unmanipulated C3H.SW donors”.

In this experiment the CD8 cells were from unmanipulated mice. We did not sort for CD44⁺CD62L⁺ cells. In a previous section in the manuscript (page 15) we wrote that, “Although we did not sort T_N, spontaneous T_M do not mount an anti-H60 response[91]; therefore, all Tet^{H60+} cells were derived from T_N.” As rightly pointed out by Reviewer 2, in other sections of the manuscript we did call such cells “T_N” to streamline the writing and figure labeling. We thank Reviewer 2 for this suggestion, and we have added “T_N”.

9. In Fig. 10A and B, it will be better to show Tox expression levels in unstimulated naïve CD8 T cells as a negative control.

We thank Reviewer 2 for this suggestion. We now include TOX expression of CD4⁺ CD8⁺ (double positive) thymocytes as a positive control and peripheral naïve CD8⁺ cells as a negative control. These data are in Supplemental Figure 9A.

10. On page 16 the last line to page 17 the first line, the authors said “TOX expression in TetH60-negative cells in Tn-recipients was bimodal, with discrete populations of PD-1 High TOX High and PD-1 low TOX low cells (Fig. 10E). FGK45 increased the frequency of TetH60 negative PD-1 high TOX high cells in spleen, perhaps by enhancing alloreactive CD8 cell expansion.” However, Fig. 10E analyzes Tcf-1 expression (no TOX expression data). So, the Reviewer thinks that Tcf1 expression in TetH60-negative cells in Tn-recipients was bimodal. Also, Fig. 10E shows that PD-1 high cells are Tcf-1 low. Conversely, PD-1 low cells are Tcf-1 high. Please revise and clarify this part.

We regret our errors here. We did mean that TOX expression was bimodal. In the final editing of the manuscript these data were to have been moved into Supplemental Figure 8, but inadvertently this was not done. It is true that TOX expression was bimodal in Tet^{H60-} cells and these data are now in Supplemental Figure 9 (which was formerly Supplemental Figure 8 as in the revised manuscript there is an additional supplemental figure) as we originally intended. We apologize for this error and the confusion and we thank Reviewer 2 for alerting us to this mistake.

We agree that the TCF-1 MFI is higher in PD-1 low cells. We pointed out that this was true for Tet^{H60+} cells but failed to mention this for the Tet^{H60-} cells. We have revised the manuscript to include this point.

References

1. Choi, E.Y., et al., *Quantitative analysis of the immune response to mouse non-MHC transplantation antigens in vivo: the H60 histocompatibility antigen dominates over all others*. J Immunol, 2001. **166**(7): p. 4370-9.
2. Malarkannan, S., et al., *The molecular and functional characterization of a dominant minor H antigen, H60*. J Immunol, 1998. **161**(7): p. 3501-9.
3. Choi, E.Y., et al., *Immunodominance of H60 is caused by an abnormally high precursor T cell pool directed against its unique minor histocompatibility antigen peptide*. Immunity, 2002. **17**(5): p. 593-603.
4. Malarkannan, S., et al., *Differences that matter: major cytotoxic T cell-stimulating minor histocompatibility antigens*. Immunity, 2000. **13**(3): p. 333-44.
5. Choi, E.Y., et al., *Real-time T-cell profiling identifies H60 as a major minor histocompatibility antigen in murine graft-versus-host disease*. Blood, 2002. **100**(13): p. 4259-65.
6. Wettstein, P.J., *Immunodominance in the T-cell response to multiple non-H-2 histocompatibility antigens. II. Observation of a hierarchy among dominant antigens*. Immunogenetics, 1986. **24**(1): p. 24-31.
7. Nevala, W.K. and P.J. Wettstein, *The preferential cytolytic T lymphocyte response to immunodominant minor histocompatibility antigen peptides*. Transplantation, 1996. **62**(2): p. 283-91.
8. Wettstein, P.J., *Immunodominance in the T cell response to multiple non-H-2 histocompatibility antigens. III. Single histocompatibility antigens dominate the male antigen*. J Immunol, 1986. **137**(7): p. 2073-9.
9. Wettstein, P.J., *Minor Histocompatibility Loci*, in *Human Immunogenetics*, S. Litwin, Editor. 1989, Marcel Dekker, Inc.: New York. p. 339-358.
10. Wettstein, P.J. and D.W. Bailey, *Immunodominance in the immune response to "multiple" histocompatibility antigens*. Immunogenetics, 1982. **16**(1): p. 47-58.
11. Korngold, R. and P.J. Wettstein, *Immunodominance in the graft-vs-host disease T cell response to minor histocompatibility antigens*. J Immunol, 1990. **145**(12): p. 4079-88.
12. Wettstein, P.J. and R. Korngold, *T cell subsets required for in vivo and in vitro responses to single and multiple minor histocompatibility antigens*. Transplantation, 1992. **54**(2): p. 296-307.
13. Berger, M., P.J. Wettstein, and R. Korngold, *T cell subsets involved in lethal graft-versus-host disease directed to immunodominant minor histocompatibility antigens*. Transplantation, 1994. **57**(7): p. 1095-102.
14. Dossa, R.G., et al., *Development of T-cell immunotherapy for hematopoietic stem cell transplantation recipients at risk of leukemia relapse*. Blood, 2018. **131**(1): p. 108-120.
15. Bleakley, M. and S.R. Riddell, *Exploiting T cells specific for human minor histocompatibility antigens for therapy of leukemia*. Immunology and cell biology, 2011.
16. Bleakley, M., et al., *Leukemia-associated minor histocompatibility antigen discovery using T-cell clones isolated by in vitro stimulation of naive CD8+ T cells*. Blood, 2010. **115**(23): p. 4923.
17. Bleakley, M. and S.R. Riddell, *Molecules and mechanisms of the graft-versus-leukaemia effect*. Nat Rev Cancer, 2004. **4**(5): p. 371-80.
18. van der Lee, D.I., et al., *Mutated nucleophosmin 1 as immunotherapy target in acute myeloid leukemia*. J Clin Invest, 2019. **129**(2): p. 774-785.
19. Van Bergen, C.A.M., et al., *High-Throughput Characterization of 10 New Minor Histocompatibility Antigens by Whole Genome Association Scanning*. Cancer Research, 2010. **70**(22): p. 9073.
20. Stumpf, A.N., et al., *Identification of 4 new HLA-DR-restricted minor histocompatibility antigens as hematopoietic targets in antitumor immunity*. Blood, 2009. **114**(17): p. 3684-3692.
21. Norde, W.J., et al., *Myeloid leukemic progenitor cells can be specifically targeted by minor histocompatibility antigen LRH-1-reactive cytotoxic T cells*. Blood, 2009. **113**(10): p. 2312-2323.
22. Falkenburg, J.H., et al., *Minor histocompatibility antigens in human stem cell transplantation*. Exp Hematol, 2003. **31**(9): p. 743-51.
23. Vogt, M.H., et al., *DFFRY codes for a new human male-specific minor transplantation antigen involved in bone marrow graft rejection*. Blood, 2000. **95**(3): p. 1100-5.
24. Tawara, I., et al., *Safety and persistence of WT1-specific T-cell receptor gene-transduced lymphocytes*

- in patients with AML and MDS*. Blood, 2017. **130**(18): p. 1985-1994.
25. Chapuis, A.G., et al., *T cell receptor gene therapy targeting WT1 prevents acute myeloid leukemia relapse post-transplant*. Nat Med, 2019. **25**(7): p. 1064-1072.
 26. Chapuis, A.G., et al., *Transferred WT1-reactive CD8+ T cells can mediate antileukemic activity and persist in post-transplant patients*. Sci Transl Med, 2013. **5**(174): p. 174ra27.
 27. Spaapen, R.M., et al., *Rapid identification of clinical relevant minor histocompatibility antigens via genome-wide zygosity-genotype correlation analysis*. Clin Cancer Res, 2009. **15**(23): p. 7137-43.
 28. Laumont, C.M., et al., *Global proteogenomic analysis of human MHC class I-associated peptides derived from non-canonical reading frames*. Nat Commun, 2016. **7**: p. 10238.
 29. Granados, D.P., et al., *Proteogenomic-based discovery of minor histocompatibility antigens with suitable features for immunotherapy of hematologic cancers*. Leukemia, 2016. **30**(6): p. 1344-54.
 30. Fontaine, P., et al., *Adoptive transfer of minor histocompatibility antigen-specific T lymphocytes eradicates leukemia cells without causing graft-versus-host disease.[comment]*. Nature Medicine., 2001. **7**(7): p. 789-94.
 31. Pion, S., et al., *Immunodominant minor histocompatibility antigens expressed by mouse leukemic cells can serve as effective targets for T cell immunotherapy*. Journal of Clinical Investigation, 1995. **95**(4): p. 1561-8.
 32. Falkenburg, F., et al., *Prevention and treatment of relapse after stem cell transplantation by cellular therapies*. Bone Marrow Transplant, 2019. **54**(1): p. 26-34.
 33. Lansford, J.L., et al., *Computational modeling and confirmation of leukemia-associated minor histocompatibility antigens*. Blood Adv, 2018. **2**(16): p. 2052-2062.
 34. Yao, C., et al., *Single-cell RNA-seq reveals TOX as a key regulator of CD8(+) T cell persistence in chronic infection*. Nat Immunol, 2019. **20**(7): p. 890-901.
 35. Khan, O., et al., *TOX transcriptionally and epigenetically programs CD8(+) T cell exhaustion*. Nature, 2019. **571**(7764): p. 211-218.
 36. Chen, Z., et al., *TCF-1-Centered Transcriptional Network Drives an Effector versus Exhausted CD8 T Cell-Fate Decision*. Immunity, 2019. **51**(5): p. 840-855 e5.
 37. Pircher, H., et al., *Tolerance induction in double specific T-cell receptor transgenic mice varies with antigen*. Nature, 1989. **342**(6249): p. 559-61.
 38. Oldstone, M.B., et al., *Virus infection triggers insulin-dependent diabetes mellitus in a transgenic model: role of anti-self (virus) immune response*. Cell, 1991. **65**(2): p. 319-31.
 39. Barber, D.L., et al., *Restoring function in exhausted CD8 T cells during chronic viral infection*. Nature, 2006. **439**(7077): p. 682-7.
 40. Wherry, E.J., et al., *Molecular signature of CD8+ T cell exhaustion during chronic viral infection*. Immunity, 2007. **27**(4): p. 670-84.
 41. Im, S.J., et al., *Defining CD8+ T cells that provide the proliferative burst after PD-1 therapy*. Nature, 2016. **537**(7620): p. 417-421.
 42. Ochsenbein, A.F., et al., *Roles of tumour localization, second signals and cross priming in cytotoxic T-cell induction*. Nature, 2001. **411**(6841): p. 1058-64.
 43. Magen, A., et al., *Single-Cell Profiling Defines Transcriptomic Signatures Specific to Tumor-Reactive versus Virus-Responsive CD4(+) T Cells*. Cell Rep, 2019. **29**(10): p. 3019-3032 e6.
 44. West, E.E., et al., *Tight regulation of memory CD8(+) T cells limits their effectiveness during sustained high viral load*. Immunity, 2011. **35**(2): p. 285-98.
 45. Penalzoza-MacMaster, P., et al., *Vaccine-elicited CD4 T cells induce immunopathology after chronic LCMV infection*. Science, 2015. **347**(6219): p. 278-82.
 46. Hale, J.S., et al., *Distinct memory CD4+ T cells with commitment to T follicular helper- and T helper 1-cell lineages are generated after acute viral infection*. Immunity, 2013. **38**(4): p. 805-17.
 47. Snook, J.P., C. Kim, and M.A. Williams, *TCR signal strength controls the differentiation of CD4(+) effector and memory T cells*. Sci Immunol, 2018. **3**(25).
 48. Grunig, G., et al., *Requirement for IL-13 independently of IL-4 in experimental asthma [see comments]*. Science, 1998. **282**(5397): p. 2261-3.
 49. Masopust, D., et al., *Preferential localization of effector memory cells in nonlymphoid tissue*. Science, 2001. **291**(5512): p. 2413-7.
 50. Reinhardt, R.L., et al., *Visualizing the generation of memory CD4 T cells in the whole body*. Nature,

2001. **410**(6824): p. 101-5.
51. Le Bon, A., et al., *Cross-priming of CD8+ T cells stimulated by virus-induced type I interferon*. Nat Immunol, 2003. **4**(10): p. 1009-15.
 52. Ballesteros-Tato, A., et al., *Temporal changes in dendritic cell subsets, cross-priming and costimulation via CD70 control CD8(+) T cell responses to influenza*. Nat Immunol, 2010. **11**(3): p. 216-24.
 53. DuPage, M., et al., *Endogenous T cell responses to antigens expressed in lung adenocarcinomas delay malignant tumor progression*. Cancer Cell, 2011. **19**(1): p. 72-85.
 54. Curran, M.A., et al., *PD-1 and CTLA-4 combination blockade expands infiltrating T cells and reduces regulatory T and myeloid cells within B16 melanoma tumors*. Proc Natl Acad Sci U S A, 2010. **107**(9): p. 4275-80.
 55. McLachlan, J.B., et al., *Dendritic Cell Antigen Presentation Drives Simultaneous Cytokine Production by Effector and Regulatory T Cells in Inflamed Skin*. Immunity, 2009. **30**(2): p. 277-288.
 56. Van Rijt, L.S., et al., *In vivo depletion of lung CD11c⁺ dendritic cells during allergen challenge abrogates the characteristic features of asthma*. Journal of Experimental Medicine, 2005. **201**(6): p. 981-991.
 57. Buchholz, V.R., et al., *Disparate individual fates compose robust CD8+ T cell immunity*. Science, 2013. **340**(6132): p. 630-5.
 58. Stemmerger, C., et al., *A single naive CD8+ T cell precursor can develop into diverse effector and memory subsets*. Immunity, 2007. **27**(6): p. 985-97.
 59. Cheung, A.F., et al., *Regulated expression of a tumor-associated antigen reveals multiple levels of T-cell tolerance in a mouse model of lung cancer*. Cancer Res, 2008. **68**(22): p. 9459-68.
 60. Deng, L., et al., *STING-Dependent Cytosolic DNA Sensing Promotes Radiation-Induced Type I Interferon-Dependent Antitumor Immunity in Immunogenic Tumors*. Immunity, 2014. **41**(5): p. 843-52.
 61. Sivan, A., et al., *Commensal Bifidobacterium promotes antitumor immunity and facilitates anti-PD-L1 efficacy*. Science, 2015. **350**(6264): p. 1084-9.
 62. Spranger, S., R. Bao, and T.F. Gajewski, *Melanoma-intrinsic beta-catenin signalling prevents anti-tumour immunity*. Nature, 2015. **523**(7559): p. 231-5.
 63. Koyama, M., et al., *Donor colonic CD103+ dendritic cells determine the severity of acute graft-versus-host disease*. J Exp Med, 2015. **212**(8): p. 1303-21.
 64. Koyama, M., et al., *MHC Class II Antigen Presentation by the Intestinal Epithelium Initiates Graft-versus-Host Disease and Is Influenced by the Microbiota*. Immunity, 2019. **51**(5): p. 885-898 e7.
 65. Alexander, K.A., et al., *CSF-1-dependant donor-derived macrophages mediate chronic graft-versus-host disease*. J Clin Invest, 2014. **124**(10): p. 4266-80.
 66. Koyama, M., et al., *Recipient nonhematopoietic antigen-presenting cells are sufficient to induce lethal acute graft-versus-host disease*. Nat Med, 2012. **18**(1): p. 135-42.
 67. Scott, A.C., et al., *TOX is a critical regulator of tumour-specific T cell differentiation*. Nature, 2019. **571**(7764): p. 270-274.
 68. Schietinger, A., et al., *Tumor-Specific T Cell Dysfunction Is a Dynamic Antigen-Driven Differentiation Program Initiated Early during Tumorigenesis*. Immunity, 2016. **45**(2): p. 389-401.
 69. Huang, C.T., et al., *Role of LAG-3 in regulatory T cells*. Immunity, 2004. **21**(4): p. 503-13.
 70. Cui, Y., et al., *Immunotherapy of established tumors using bone marrow transplantation with antigen gene-modified hematopoietic stem cells*. Nat Med, 2003. **9**(7): p. 952-958.
 71. Sotomayor, E.M., et al., *Cross-presentation of tumor antigens by bone marrow-derived antigen-presenting cells is the dominant mechanism in the induction of T-cell tolerance during B-cell lymphoma progression*. Blood, 2001. **98**(4): p. 1070-7.
 72. Juchem, K.W., et al., *PD-L1 Prevents the Development of Autoimmune Heart Disease in Graft-versus-Host Disease*. J Immunol, 2018. **200**(2): p. 834-846.
 73. Juchem, K.W., et al., *A repertoire-independent and cell-intrinsic defect in murine GVHD induction by effector memory T cells*. Blood, 2011. **118**(23): p. 6209-19.
 74. Ohlen, C., et al., *CD8(+) T cell tolerance to a tumor-associated antigen is maintained at the level of expansion rather than effector function*. J Exp Med, 2002. **195**(11): p. 1407-18.
 75. Stromnes, I.M., et al., *Abrogating Cbl-b in effector CD8(+) T cells improves the efficacy of adoptive therapy of leukemia in mice*. J Clin Invest, 2010. **120**(10): p. 3722-34.
 76. Korn, T., et al., *Myelin-specific regulatory T cells accumulate in the CNS but fail to control autoimmune*

- inflammation*. Nat Med, 2007. **13**(4): p. 423-31.
77. Chen, Y., et al., *Anti-IL-23 therapy inhibits multiple inflammatory pathways and ameliorates autoimmune encephalomyelitis*. J Clin Invest, 2006. **116**(5): p. 1317-26.
78. Yan, S.S., et al., *Suppression of experimental autoimmune encephalomyelitis by selective blockade of encephalitogenic T-cell infiltration of the central nervous system*. Nat Med, 2003. **9**(3): p. 287-93.
79. Rojas, O.L., et al., *Recirculating Intestinal IgA-Producing Cells Regulate Neuroinflammation via IL-10*. Cell, 2019. **176**(3): p. 610-624 e18.
80. Jordao, M.J.C., et al., *Single-cell profiling identifies myeloid cell subsets with distinct fates during neuroinflammation*. Science, 2019. **363**(6425).
81. Hogquist, K.A., et al., *T cell receptor antagonist peptides induce positive selection*. Cell, 1994. **76**(1): p. 17-27.
82. Butz, E.A. and M.J. Bevan, *Massive expansion of antigen-specific CD8+ T cells during an acute virus infection*. Immunity, 1998. **8**(2): p. 167-75.
83. Utzschneider, D.T., et al., *T Cell Factor 1-Expressing Memory-like CD8(+) T Cells Sustain the Immune Response to Chronic Viral Infections*. Immunity, 2016. **45**(2): p. 415-27.
84. He, B., et al., *CD8(+) T Cells Utilize Highly Dynamic Enhancer Repertoires and Regulatory Circuitry in Response to Infections*. Immunity, 2016. **45**(6): p. 1341-1354.
85. Bengsch, B., et al., *Bioenergetic Insufficiencies Due to Metabolic Alterations Regulated by the Inhibitory Receptor PD-1 Are an Early Driver of CD8(+) T Cell Exhaustion*. Immunity, 2016. **45**(2): p. 358-73.
86. Cui, W., et al., *An interleukin-21-interleukin-10-STAT3 pathway is critical for functional maturation of memory CD8+ T cells*. Immunity, 2011. **35**(5): p. 792-805.
87. Rutishauser, R.L., et al., *Transcriptional repressor Blimp-1 promotes CD8(+) T cell terminal differentiation and represses the acquisition of central memory T cell properties*. Immunity, 2009. **31**(2): p. 296-308.
88. Shlomchik, W.D., et al., *Prevention of graft versus host disease by inactivation of host antigen-presenting cells*. Science, 1999. **285**(5426): p. 412-5.
89. Wang, X., et al., *Mechanisms of antigen presentation to T cells in murine graft-versus-host disease: cross-presentation and the appearance of cross-presentation*. Blood, 2011. **118**(24): p. 6426-37.
90. Markey, K.A., et al., *Conventional dendritic cells are the critical donor APC presenting alloantigen after experimental bone marrow transplantation*. Blood, 2009. **113**(22): p. 5644-9.
91. Li, N., et al., *Memory T cells from minor histocompatibility antigen-vaccinated and virus-immune donors improve GVL and immune reconstitution*. Blood, 2011. **118**(22): p. 5965-76.
92. Dash, A.B., et al., *A murine model of CML blast crisis induced by cooperation between BCR/ABL and NUP98/HOXA9*. Proc Natl Acad Sci U S A, 2002. **99**(11): p. 7622-7.
93. Huntly, B.J. and D.G. Gilliland, *Leukaemia stem cells and the evolution of cancer-stem-cell research*. Nat Rev Cancer, 2005. **5**(4): p. 311-21.
94. Frohling, S., et al., *Genetics of myeloid malignancies: pathogenetic and clinical implications.[see comment]*. Journal of Clinical Oncology, 2005. **23**(26): p. 6285-95.
95. Patel, J.P., et al., *Prognostic relevance of integrated genetic profiling in acute myeloid leukemia*. N Engl J Med, 2012. **366**(12): p. 1079-89.
96. Sun, Y., et al., *HOXA9 Reprograms the Enhancer Landscape to Promote Leukemogenesis*. Cancer Cell, 2018. **34**(4): p. 643-658 e5.
97. Wilkinson, A.N., et al., *IL-6 dysregulation originates in dendritic cells and mediates graft-versus-host disease via classical signaling*. Blood, 2019. **134**(23): p. 2092-2106.
98. Gartlan, K.H., et al., *Donor T-cell-derived GM-CSF drives alloantigen presentation by dendritic cells in the gastrointestinal tract*. Blood Adv, 2019. **3**(19): p. 2859-2865.
99. Markey, K.A., et al., *Fit-3L Expansion of Recipient CD8alpha(+) Dendritic Cells Deletes Alloreactive Donor T Cells and Represents an Alternative to Posttransplant Cyclophosphamide for the Prevention of GVHD*. Clin Cancer Res, 2018. **24**(7): p. 1604-1616.
100. Bruce, D.W., et al., *Type 2 innate lymphoid cells treat and prevent acute gastrointestinal graft-versus-host disease*. J Clin Invest, 2017. **127**(5): p. 1813-1825.
101. Zhang, P., et al., *Induced regulatory T cells promote tolerance when stabilized by rapamycin and IL-2 in vivo*. J Immunol, 2013. **191**(10): p. 5291-303.
102. Ni, X., et al., *PD-L1 interacts with CD80 to regulate graft-versus-leukemia activity of donor CD8+ T*

- cells. *J Clin Invest*, 2017. **127**(5): p. 1960-1977.
103. Matte-Martone, C., et al., *Differential requirements for myeloid leukemia IFN-gamma conditioning determine graft-versus-leukemia resistance and sensitivity*. *J Clin Invest*, 2017. **127**(7): p. 2765-2776.

REVIEWERS' COMMENTS:

Reviewer #2 (Remarks to the Author):

The authors answered all concerns/issues raised by the reviewer.